# Towards silent and efficient flight by combining bioinspired owl feather serrations with cicada wing geometry

Zixiao Wei [1,2], Stanley Wang[1,2], Sean Farris[1], Naga Chennuri [1], Ningping Wang[1], Stara Shinsato[1], Kahraman Demir[1], Maya Horii [1] & Grace X. Gu [1] ✉

As natural predators, owls fly with astonishing stealth due to the serrated feather morphology that produces advantageous flow characteristics. Traditionally, these serrations are tailored for airfoil edges with simple two-dimensional patterns, limiting their effect on noise reduction while negotiating tradeoffs in aerodynamic performance. Conversely, the intricately structured wings of cicadas have evolved for effective flapping, presenting a potential blueprint for alleviating these aerodynamic limitations. In this study, we formulate a synergistic design strategy that harmonizes noise suppression with aerodynamic efficiency by integrating the geometrical attributes of owl feathers and cicada forewings, culminating in a three-dimensional sinusoidal serration propeller topology that facilitates both silent and efficient flight. Experimental results show that our design yields a reduction in overall sound pressure levels by up to 5.5 dB and an increase in propulsive efficiency by over 20% compared to the current industry benchmark. Computational fluid dynamics simulations validate the efficacy of the bioinspired design in augmenting surface vorticity and suppressing noise generation across various flow regimes. This topology can advance the multifunctionality of aerodynamic surfaces for the development of quieter and more energy-saving aerial vehicles.

Smart aeroacoustics design draws great attention due to the rapid popularization of urban usage of aerial vehicles and rising restrictions on noise pollution. In the conventional framework of propeller design, B-spline methodologies[1] play a pivotal role. This involves generating a series of control points through basis functions, which are instrumental in formulating the aerodynamic surface. Nevertheless, the interplay between aerodynamic efficiency and noise reduction is delicate, and finding a balance often requires stepping outside established design paradigms. Researchers are alternatively exploring the flight mechanisms of natural predators renowned for their stealth, aiming to derive innovative and efficacious design principles. Biological entities, having undergone evolutionary refinement over billions of years, offer a wealth of inspiration for developing human-engineered products and systems[2–7]. For instance, creatures like the owl have developed unique wings and feather morphologies that enhance stealth during flight[8]. This adaptation has been an essential element in the design of bioinspired passive noise control devices. Wang et al.[9] provide a review of owl wing-inspired aeroacoustic devices and identify three distinct features of an owl feather: leading-edge serrations, trailing-edge fringes, and a soft downy (velvet-like) coating. Owl feather-inspired modifications to aerodynamic surfaces have resulted in a variety of solutions, including variations of leading-edge serrations[10–13], trailing-edge serrations[14], and the application of porous dampening materials[15]. Yet, in conventional designs using

---

[1]Department of Mechanical Engineering, University of California, Berkeley, CA, USA. [2]These authors contributed equally: Zixiao Wei, Stanley Wang.
✉e-mail: ggu@berkeley.edu

two-dimensional (2D) patterns such as the leading-edge sawtooth serration, slitted serration[16], and sinusoidal serration, the pursuit of passive noise reduction often comes at a penalty of overall aerodynamic performance. In this context, the sophisticated shape of cicada forewings has garnered interest due to their exceptional aerodynamic features. Despite the difference in Reynolds number (Re) between the flight dynamics of owls[17] (Re≈6.0e4) and cicadas[18] (Re≈2.0e3), existing research from numerical simulations[18–20] and empirical studies[21] suggests that cicada wing planforms offer aerodynamic benefits across a wide range of Re, including those at which conventional drone propellers operate[22,23].

In this work, we formulate new design strategies that can mitigate tradeoffs between noise reduction and aerodynamic performance by merging owl feather and cicada wing geometries to create a propeller topology that features silent and efficient flights. Inspired by the morphology of owl feathers, our design introduces a high-fidelity, three-dimensional (3D) sinusoidal serration topography that encompasses a widespread surface adaptation rather than a localized edge variation for potential acoustic improvement, as illustrated in Fig. 1a. Integrating this design, the cicada wings' contour serves as the foundational propeller planform to augment aerodynamic efficiency. To validate this synergistic design strategy, a hybrid aeroacoustic and aerodynamic measurement system is employed in our experiments. A group of propellers consisting of several representative designs is examined and compared to systematically discern and evaluate the individual and combined influences of the owl feather and cicada wing characteristics on performance. The topology highlighting our design

is the 3D sinusoidal cicada (3D-SC) topology. Our control group has four benchmark designs. A smooth, cicada-shaped prototype is set as the first benchmark (B1). A conventionally shaped, serrated prototype is the second benchmark (B2). A conventionally shaped, non-serrated prototype is the third benchmark (B3). In this context, the comparison between 3D-SC and B1 helps to delineate the influence of the 3D surface serrations. The comparison between B1 and B3 provides insights into the impact of the cicada planform. Moreover, we introduce the DJI Phantom 3 propeller as an extra benchmark (B4), establishing it as a comparative standard for contemporary, state-of-the-art industrial prototypes. These experimental prototypes are fabricated using Polyjet additive manufacturing processes discussed in the "Methods" section and as shown in Fig. 1b. Moreover, computational fluid dynamics (CFD) simulations are performed to investigate the underlying mechanisms, providing a complementary perspective to experimental findings. The fundamental mechanisms studied in this work can lead to promising applications in a multitude of fields, including urban air mobility, wind power generation, and hydrodynamic vehicles.

## Results

### Topological design concepts

In the design process for the 3D-SC propeller, we start by digitalizing the morphology of the owl fringe and cicada forewing shapes such that their geometries can be expressed explicitly and integrated smoothly as illustrated in Fig. 1a. The chord distribution function, denoted as $C$, for the cicada wing-like planform is defined along the spanwise

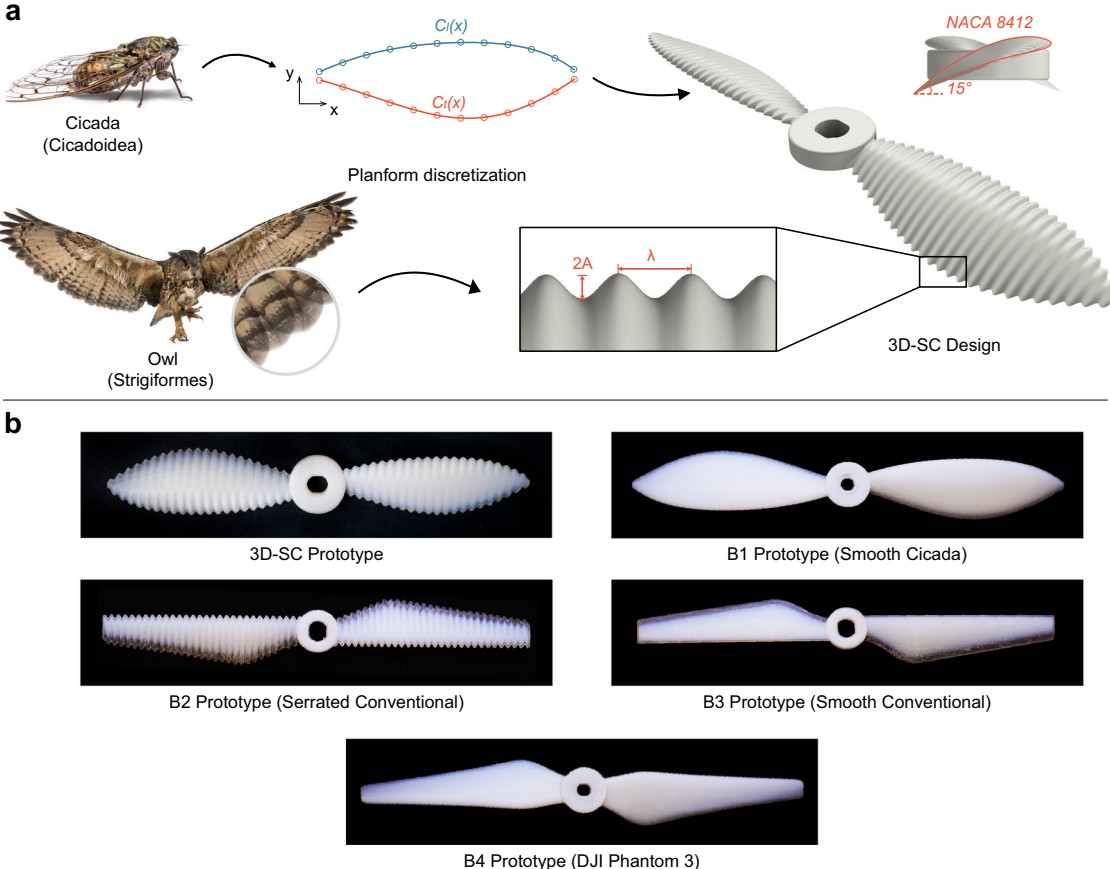

**Fig. 1 | Illustration of the 3D-SC propeller design process. a** This panel depicts the sequential development of the 3D-SC propeller topology, based on the morphological characteristics of owl feathers and the shape of cicada forewings, ending with the CAD representation on the right. The inset highlights the definitions of wavelength and amplitude of the sinusoidal serration waveform, alongside the 2D airfoil profile in the side view. Image of cicada and owl courtesy of Eric Isselee/ Shutterstock.com. **b** This panel displays the fabricated 6-inch diameter 3D-SC propeller along with the B1, B2, B3, and B4 benchmarks, produced using Polyjet additive manufacturing, illustrating the process from the design concept to the physical prototype.

position $x$, in relation to the total blade span, $b$. This distribution function is derived from fitting a fifth-order polynomial to the chord length profile of a typical cicada wing, using separate functions for the leading ($l$) and trailing ($t$) edges, as depicted in Fig. 1a. In our study, the span $b$ is set at 3 inches, which equals the rotor's radius. It is of note that the chosen cicada wing shape is a preliminary model, serving as a starting point rather than a refined design. The potential to fine-tune this structure to enhance performance metrics presents a valuable opportunity for future research, which is not covered within the present study's scope.

$$C_l(x) = -\frac{0.6455}{b^4}x^5 + \frac{0.9153}{b^3}x^4 - \frac{0.2210}{b^2}x^3 - \frac{0.5010}{b}x^2 \atop + 0.4579x + 0.01680b \tag{1}$$

$$C_t(x) = -\frac{0.6494}{b^4}x^5 + \frac{1.1472}{b^3}x^4 + \frac{0.02558}{b^2}x^3 - \frac{0.2415}{b}x^2 \atop - 0.2766x - 0.01647b \tag{2}$$

The guidance splines of 3D surface serrations are then created by superimposing a sinusoidal pattern onto the chord distribution functions for the leading and trailing edges ($C_{l/t}$), as demonstrated by Eqs. (3–4) (see Supplementary Fig. 1a–c for details). To describe the sinusoidal waveform, a dimensionless variable known as aspect ratio ($\Lambda$) is introduced, as shown in Eq. (5). In this framework, $A$ signifies the amplitude, while $\lambda$ corresponds to the wavelength of the base sinusoidal waveform denoted by $s$. Here, $\Lambda$ represents the slope, expressing how $\lambda$ varies as a function of $A$.

$$C_{3D-sc} = C_{l/t}(x) + C_s(x) \tag{3}$$

$$C_s(x) = A \cdot \sin\left(\frac{2\pi}{\lambda}x\right) \tag{4}$$

$$\lambda = \Lambda \cdot A + \lambda_0 \tag{5}$$

It is noteworthy that the sound pressure level (SPL) produced by a rotor is significantly influenced by the turbulent boundary layer (TBL) attached to the blade surface. Re is, hence, introduced to characterize the level of flow turbulence:

$$Re = \frac{\rho \omega x C}{\mu}\cos(\alpha) \tag{6}$$

where $\omega$ denotes the rotor's angular rotational speed in rad/s, and $\mu$ represents the dynamic viscosity of air. The symbol $\alpha$ is used to denote the effective angle of attack for the airfoil's cross-section. The Reynolds number corresponding to each experimental scenario is listed in Supplementary Table 1. The 2D airfoil used for lofting is NACA 8412, which features a high lift coefficient and lift to drag ratio, and stalls at approximately 25 degrees at the operating Re, as shown in Supplementary Fig. 1d. The cross-span airfoil pitch angle of the cicada-based prototypes (i.e., 3D-SC and B1) is fixed to be 15 degrees to prevent flow separation. To provide a basis for comparison, the pitch angle for each cross-section in prototypes B2 and B3 is uniformly established. The CAD model of the B4 prototype is derived from 3D scanning and rescaling processes to ensure an accurate representation of the industry benchmark.

## Acoustic comparison with 2D leading-edge serrations

Before the study of the synergistic 3D-SC topology, the acoustic performance of 2D leading-edge serrations is compared with the 3D surface serrations. In the experiment, two identical, conventionally

shaped propellers are respectively reinforced by these two types of serrations. To avoid any potential bias arising from differences in sinusoidal waveforms, we maintain a consistent amplitude and wavelength of 0.04 and 0.4 inches for the 3D-serrated prototype. The leading-edge serrated prototypes are designed with a range of amplitudes and wavelengths, encompassing the 0.04 × 0.4-inch baseline design, to enable a comprehensive comparison. Based on the experimental results, the overall sound pressure level (OASPL) of a 3D-serrated propeller is 3.63 dB lower than the leading-edge serrated propeller when their amplitudes and wavelengths are identical, as detailed in Supplementary Fig. 2a. Furthermore, the figure demonstrates that the 3D-serrated prototype results in a consistently lower acoustic emission across the spectrum than the various leading-edge serrated prototypes designed with different waveforms. This empirical finding suggests that the enhanced surface texture provided by 3D serrations achieves a further reduction in propeller noise relative to 2D serrations.

## Acoustic advancement at various experimental conditions

To examine the aeroacoustic performance of our designs, we first focus on the composition of rotor noise caused by propellers. In general, rotor noise consists of two primary sources: tonal and broadband noises[24]. These two sound sources are driven by different physical mechanisms and exhibit distinctive signatures in the sound spectrum. Specifically, tonal, or harmonic, noise is caused by periodic rotation, presenting as anchored pitches correlated to the rotation frequency. Tonal noise can be further broken down into loading noise and thickness noise. Loading noise is triggered by both steady and unsteady aerodynamic loading, while thickness noise is generated by local fluid expansion over the propeller surface. Broadband noise also has different sources of noise generation. The most dominating noise source is blade-wake interaction noise, resulting from the interaction between tip vortices and the propeller blade[25]. The mathematical formulation of these noise components is detailed in the Supplementary Note 1.

In the experimental setup, the sound data are collected using an omnidirectional microphone while the rotor is powered by a portable thrust stand (see the "Methods" section). Acoustic measurements are conducted at two distinct radial distances, 0.1 m and 5 m from the rotor, providing a near-field and a relatively more distant comparative measurement. At each radial position, the microphone is circumferentially positioned around the thrust stand to collect sound data from different angles. In the experiment, the sound spectra of each prototype, corresponding to different thrusts and radial distances, are recorded from 0 to 2 kHz. The sound spectrum is subsequently integrated to evaluate OASPL, as shown in Fig. 2b. These measurements are taken with the microphone positioned at the rotor's frontal axis (0°). The trend indicates that the OASPL of rotor noise generally increases with thrust. Regarding data uncertainty, the microphone positioned at 0.1 m records a standard deviation in OASPL ranging between 0.37 and 0.66 dB, with a maximum coefficient of variation (CV) of 0.56%. At 5 meters, the standard deviation fluctuates between 0.41 and 0.79 dB, with the maximum CV reaching 0.86%.

Experimental results show the cicada wing-inspired B1 prototype achieves up to 1.6 dB lower OASPL than the conventional B3 prototype at 0.1 m with 50 gram-forces (gf) thrust, increasing to 1.9 dB at 5 meters. By contrast, the implementation of 3D serrations carries out a more significant reduction in noise. The 3D-SC prototype showcases a noise decrease of 8.3 dB relative to B1 under identical conditions (0.1 m, 50 gf). This effect of 3D serrations is consistently observed in its modification to the conventional planform. In particular, the OASPL of B2 is 8.8 dB lower than that of B3. In the context of industry benchmarks, the OASPL of the B4 prototype is lower than the non-serrated models (B1 and B3), yet it remains higher than that of the serrated designs (3D-SC and B2). Notably, the 3D-SC prototype demonstrates a reduction of 5.1 dB when benchmarked against B4 at 5 m, 50 gf.

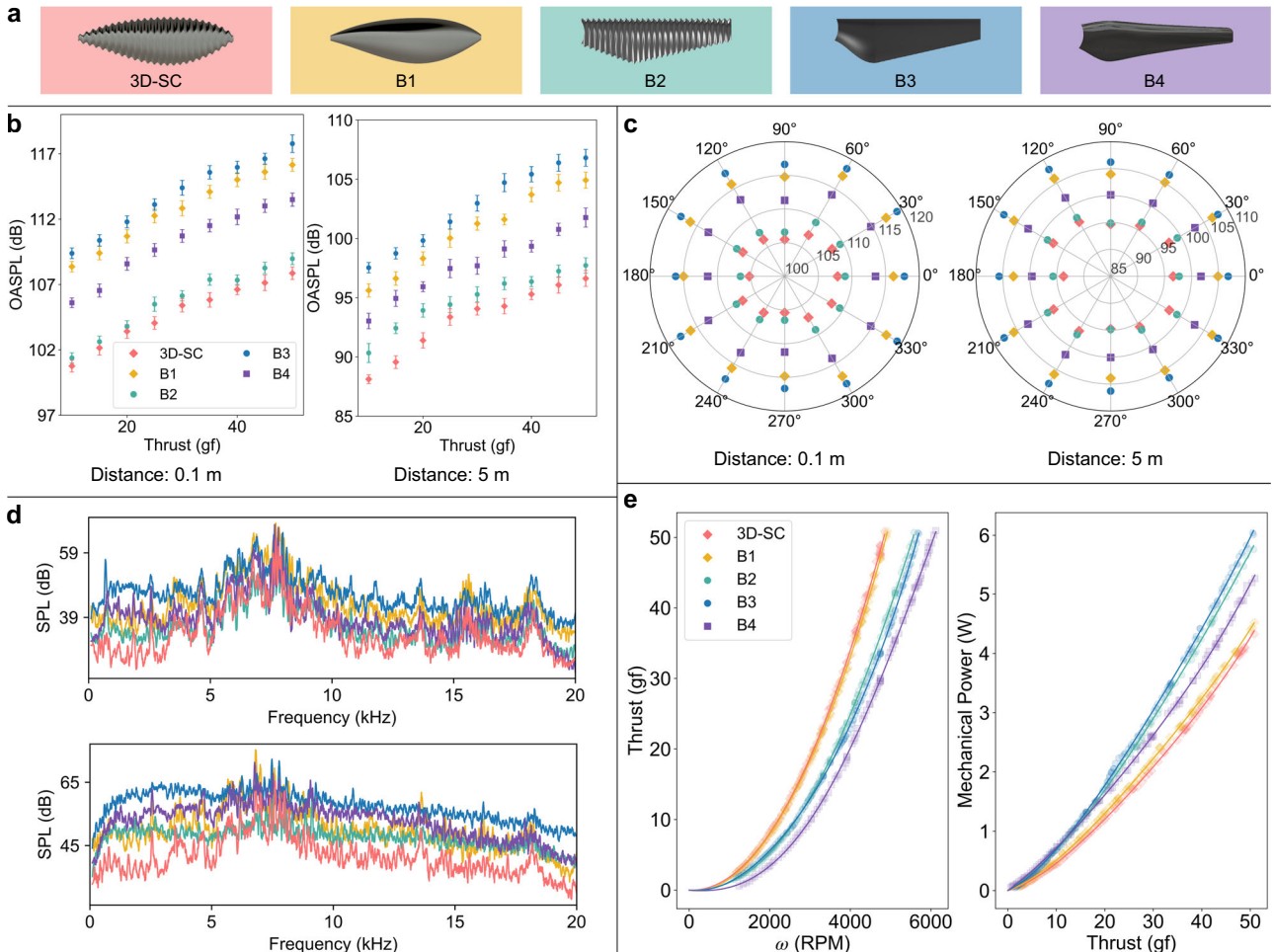

**Fig. 2 | Aerodynamic and aeroacoustic evaluation of propeller designs. a** CAD representations of the 3D-SC, B1, B2, B3, and B4 propellers are presented with their respective color codes: 3D-SC in light red, B1 in golden yellow, B2 in teal, and B4 in medium purple. **b** OASPL data across thrust values from 10 to 50 gf are displayed with error bars denoting one standard deviation of measurement variability. Each marker is positioned at the statistical mean of the corresponding data set. **c** OASPL levels are presented in polar coordinates, reflecting a range of measurement angles.

Data are collected at radial distances of 0.1 m (left panel) and 5 m (right panel) when the thrust is fixed at 50 gf. The radial axis represents the OASPL level, and the angular axis indicates the angle of measurement. **d** Comparison of acoustic spectra at a measurement distance of 0.1 m, presented for thrust levels of 15 gf (top panel) and 50 gf (bottom panel). **e** The graphs illustrate the correlation between thrust and propeller rotational speed (left), and mechanical power against thrust (right).

Figure 2c presents the sound distribution in a polar coordinate system at distances of 0.1 m and 5 m, with radial and angular coordinates representing the OASPL level and the measurement angle, respectively. A constant thrust of 50 gf is applied in this evaluation to simulate a high-thrust scenario akin to that of a drone propeller during flight. The level of noise reduction is more pronounced at the 0.1 m measurements. As the measurement distance increases, the amount of reduction declines accordingly. For a quantitative assessment of acoustic performance, a measurement distance of 5 m is selected to represent drone noise as perceived in the far-field. The peak noise abatement is achieved at a 30° measurement angle, where the 3D-SC design demonstrate a 5.5 dB noise reduction compared to the industry benchmark B4. The analyses herein utilize the unweighted OASPL. However, for industry relevance, the A-weighted OASPL—which accentuates frequencies within the human auditory range—is often studied (see Supplementary Note 1). In the A-weighted scale, the 3D-SC prototype delivers a maximum noise reduction of 5.2 dB(A) compared to the B4 design (Supplementary Fig. 2b).

To explore the effective noise attenuation bandwidth, an analysis of the rotor sound spectrum is conducted. Sound spectra at 15 gf (corresponding to Re ≈ 1.01e4) and 50 gf (corresponding to

Re ≈ 1.82e4) have been selected to highlight the acoustic characteristics of the propellers across different flow regimes, as depicted in Fig. 2d. In the spectral analysis, a 100 Hz cut-off has been applied to mitigate the influence of any constant mean bias in the signal (i.e., DC bias). To ensure that the characteristics of the investigated sound signals remain unattenuated by distance, the measurement is set at a close range of 0.1 m. The spectral analysis employs a 2nd-order Savitzky-Golay filter[26], utilizing a sliding window of 333 points to smooth the data and mitigate noise. It is observed that the rotor spectrum at 15 gf is characterized by dominant acoustic tones with frequencies below 10 kHz. This confirms that loading noise and its harmonics are the primary components of propeller noise at low rotational speeds. Notably, the SPLs of multiple tones of a 3D-SC propeller are apparently lower than B1-4. At higher Re represented by 50 gf, these SPL peaks diminish in prominence, giving way to broadband noise as the primary acoustic signature. A comparative study of the B1 and B3 prototypes reveals that the cicada wing-shaped planform alone has a limited effect on noise mitigation. Conversely, the application of 3D serrations leads to a reduction in SPL across the entire frequency spectrum, as shown by the comparative analyses between the 3D-SC and B1 prototypes, as well as between B2 and B3.

## Experimental evidence of 3D-SC aerodynamic advantages

After confirming the aeroacoustic advancement of the 3D-SC design, we proceed to probe any potential aerodynamic trade-offs associated with the observed noise reduction. In static tests of rotors[27], the thrust coefficient, $C_T = \frac{T}{(\rho n^2 d^4)}$, serves as a metric for evaluating propeller performance in thrust generation. Here, $T$ denotes thrust production, $\rho$ represents the freestream fluid density, $n$ is the rotational speed in revolutions per second, and $d$ signifies the rotor diameter. In terms of aerodynamic efficiency, the mechanical power, $W = Q \cdot n$, of a propeller is a critical determinant of energy consumption rates during flight, where $Q$ is the propeller torque.

As depicted in Fig. 2e, the thrust generation of propellers is assessed across rotational speeds up to 6000 revolutions per minute (RPM), with average mechanical power measured at thrust levels from 10 to 50 gf. Thrust measurement variability is characterized by standard deviations ranging from 0.29 gf to 0.66 gf, with a maximum CV of 5.34% at the 10-gf thrust level and 1.0% at 50 gf. Concerning mechanical power measurements, the predominant source of uncertainty originates from torque acquisition. Power fluctuations have a standard deviation between 0.032 to 0.097 W, yielding a maximum CV of 12 % at 10 gf and 1.7% at 50 gf. Detailed quantification of the aerodynamic data uncertainty across various thrust levels is delineated in Supplementary Fig. 3.

A thrust enhancement of 14.8 gf, or 39.2%, is observed in the B1 versus B3 comparison at 5000 RPM, credited to the implementation of the cicada wing planform. Furthermore, the 3D-SC propeller outperforms other models across all speeds, notably at higher RPMs. In particular, the 3D-SC design produces an extra 20.3 gf of thrust over B4 at 5000 RPM. Analysis of the thrust coefficient reveals that the 3D-SC prototype exhibits a coefficient that is approximately 0.04 higher than that of the B4 model, representing an enhancement of 55.6% (Supplementary Fig. 2c).

Assessments of propulsive efficiency suggest that the 3D-SC design not only boosts thrust but also diminishes power consumption, thus enhancing propulsive efficiency. When assessed at an equivalent thrust of 50 gf, the 3D-SC prototype exhibits a reduction in mechanical power by 0.17 W, an improvement of 4.1% compared to the B1 design, indicating the marginal efficiency benefits conferred by 3D serrations. In contrast, the B1 versus B3 comparison highlights a significant efficiency leap with the cicada wing planform, achieving a 1.49 W power reduction, equating to a 26.7% decrease. Compared to the B4 design, the 3D-SC model shows a 20.2% reduction in power consumption, indicative of concurrent improvements in thrust production and energy efficiency. Confined by the limited strength of digital ABS material, the rotor diameter is set as 6 inches, allowing for the assessment of all prototypes across a wider spectrum of rotational speeds without causing significant elastic deformation. This dimension is notably smaller than the propellers commonly employed in drones[27]. To investigate the scalability of our findings for larger-scale applications, additional experiments are performed on 12-inch 3D-SC and B4 propellers, operating at a maximum rotational speed of 3000 RPM. The outcomes of these tests are detailed in Supplementary Fig. 4, enabling the assessment of propeller performance at elevated thrust levels. At an equivalent thrust of 150 gf, the 3D-SC propeller achieves a mechanical power reduction of 2.29 W, corresponding to an improvement of 22.6% over the B4 prototype.

## Parametric study of constitutive sinusoidal wave functions

The noise attenuation effect observed in the 3D-SC topology is primarily attributed to its 3D surface serrations. Consequently, the formulation of the sinusoidal waveform is a critical determinant of the resulting performance metrics. Accordingly, we conducted a parametric optimization to produce a greater improvement in acoustic emission for a 3D-SC propeller. In this investigation, sixteen combinations of amplitude and wavelength are investigated. As shown in Fig. 3a, the testing matrix consists of different sinusoidal patterns with amplitudes varying from 0.01 to 0.04 inches and wavelengths ranging from 0.1 to 0.4 inches, representing the transition from smooth to densely serrated surface. The experimental data of thrust and OASPL for these propellers are collected from 2000 to 6000 RPM. Moreover, a high-order surface interpolation is employed to construct the contour plots of OASPL and thrust in relation to the design parameters, as shown in Fig. 3b.

At 2000 RPM, the 3D-SC propeller achieves the lowest OASPL at a wavelength of 0.1 inches and an amplitude of 0.04 inches. As demonstrated in Fig. 3a, this prototype is constructed with the densest serrations. The same waveform, however, results in high OASPL at 5000 RPM. This opposite trend implies that an excessively dense serration pattern can reduce noise at low Re while augmenting it at high Re. Furthermore, diagonal equipotential lines manifest within these contour plots, along which the rate of change in amplitude relative to wavelength remains constant. The magnitudes of OASPL and thrust corresponding to the data points situated on an equipotential line demonstrate a close numerical proximity. Particularly, we find that the properties of a 3D-SC propeller are determined by the slope (i.e., the aspect ratio $\Lambda$) of an equipotential line and its intersection point with the $y$-axis (i.e., $\lambda_0$). Specifically, a local minimum in OASPL of 79.7 dB is achieved at 2000 RPM with parameters $\Lambda = 16.0$ and $\lambda_0 = -0.223$. At an increase speed of 5000 RPM, the OASPL reaches a local minimum of 91.4 dB, at $\Lambda = 16.2$ and $\lambda_0 = 0.073$. Regarding thrust generation, the contour reveals that designs exhibiting a reduced degree of serration, distinguished by increased wavelengths and diminished wave amplitudes, are generally linked with a higher production of thrust.

To this end, our empirical results suggest an improvement in both aerodynamic and aeroacoustic performance when combining serration and planform optimization in the 3D-SC design. Moreover, the aerodynamic and acoustic characteristics of the 3D-SC propeller exhibit a correlation with its constitutive waveform parameters. In pursuit of a deeper comprehension of the mechanism underlying these observations, we utilize CFD simulations to investigate the fluid dynamics associated with the 3D-SC topology in the next section.

## Analysis of vortex manipulation mechanisms for rotor noise mitigation

CFD simulations are utilized to model the rotational dynamics of propellers within specified physical constraints. For the aerodynamic analysis, Large Eddy Simulation (LES) is employed as the numerical solver to capture the turbulent flow characteristics. Concurrently, the Ffowcs Williams-Hawkings (FW-H) method is adopted for acoustic analysis. This integrated simulation approach ensures that, at each time step, the velocity and pressure fields solved by the aerodynamic solver are interfaced with the acoustic solver, enabling the determination of the resultant sound pressures. Details regarding the formulization of acoustic and aerodynamic solvers are respectively delineated in Supplementary Note 1 and 2. In the context of rotor noise composition, the turbulence level is of great importance in analyzing the characteristics of the sound spectrum and fundamental noise reduction mechanism. Therefore, we conduct simulations at both low (2000 RPM) and high (5000 RPM) rotational speeds to examine the rotor under varying flow regimes, specifically distinguishing between laminar and turbulent flow conditions.

Based on streamline plots, shown in Fig. 4a, the serration-reinforced surface exhibits two distinctive flow features compared to a smooth surface. The first is a spanwise, centrifugal airstream, and the second is vortices clustering near the tip and the trailing edge of the blade. The identification of these prominent vortex structures is essential to understanding the interplay between aerodynamics and the resultant acoustic effect. To facilitate this analysis, iso-helicity[28] and swirling strength contours are utilized to assist in the

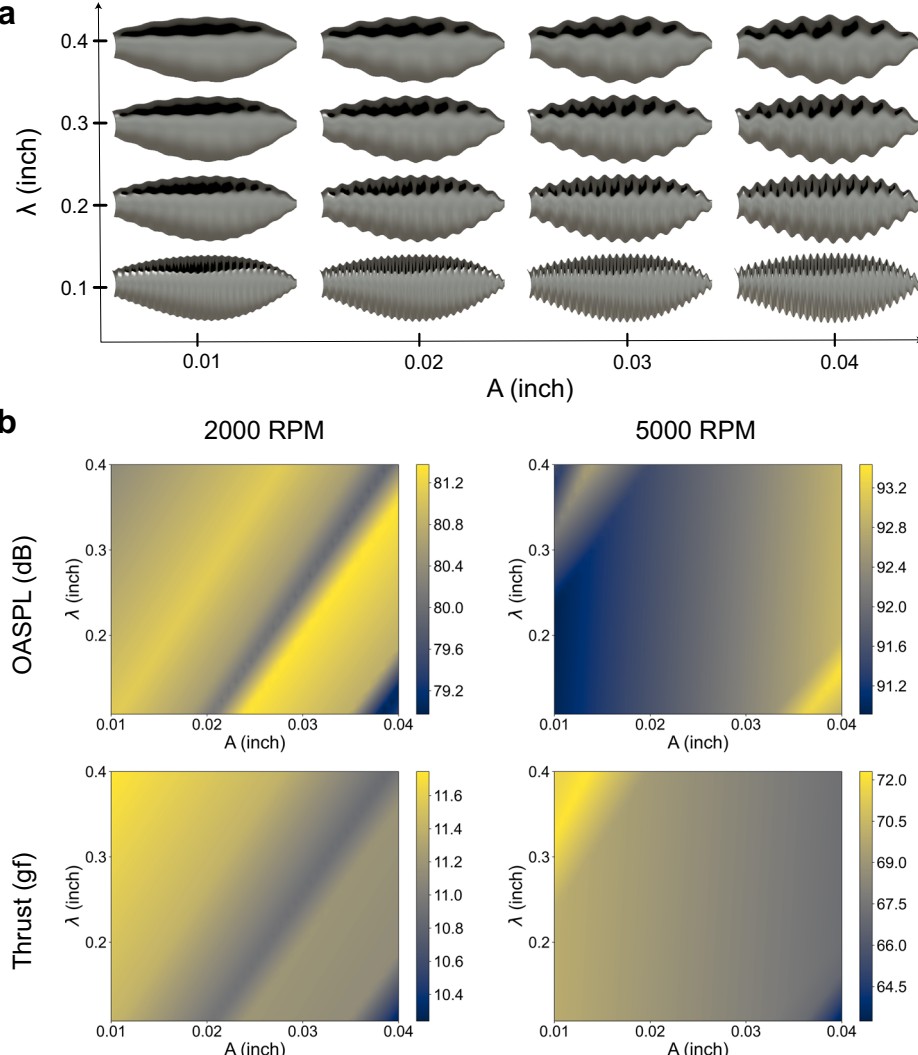

**Fig. 3 | Parametric optimization of 3D-SC propeller designs with sinusoidal elements. a** This figure visualizes a range of 3D-SC prototype designs incorporating sinusoidal serrations with variable parameters. Amplitudes (A) of these sinusoidal components extend from 0.01 to 0.04 inches, and the wavelengths (λ) from 0.1 to 0.4 inches. **b** Cividis color-coded contour plots represent the performance outcomes for the prototypes, detailing OASPL in the top panel and thrust in the bottom panel, across rotational speeds of 2000 RPM and 5000 RPM.

visualization of the vortex field. As an invariant of Euler's equation, helicity is defined as:

$$H = \int_V \mathbf{u} \cdot (\nabla \times \mathbf{u})\, dV \qquad (7)$$

where $\nabla \times u$ stands for the curl of the velocity field, while the volume integration quantifies the total amount of rotation within the controlled volume that encloses vortex lines. From the standpoint of topological fluid dynamics, helicity also reflects the linkage and knottedness of vortex lines[29–31]. In terms of a flow field within a turbulent flow regime, the eigenvalues of its velocity gradient tensor $\nabla u$ consist of one real eigenvalue and two complex conjugate eigenvalues $(\lambda_r, \lambda_c \pm \lambda_{ci} i)$. The swirling strength is defined by the imaginary part $\lambda_{ci}$, which represents the intensity of the local rotational motion.

According to Fig. 4b, the 3D-SC propeller contributes to full-span helicity at 2000 RPM (Re ≈ 7.5e3). In contrast, the surface flow field around the B1 prototype is vortex-sparse, with isolated vortices manifesting near the propeller's tip. The swirling strength iso-surface, as depicted in Fig. 4c, illustrates an analogous vortex arrangement in which the 3D-SC prototype generates vortex structures that are more

coherent and larger in scale. Particularly at the trailing edge region of the propeller, the 3D-SC prototype distinctly generates consistent streamwise vortices, a feature not observed on a smooth surface. Additionally, the swirling strength iso-surface is color-coded by the local flow vorticity. The surface vorticity distribution indicates that 3D serrations promote swifter rotational flow motions than those observed over a smooth surface. The interaction between laminar flow and a smooth array wall induces steady pressure variation at each charge of loading, which causes the growth of dipole pressure sources (i.e., tonal noises). In this sense, smoothness is not desired for tonal noise reduction at low Re. On the contrary, the valley-like, serrated surface configuration delivers a localized pressure difference (see Supplementary Fig. 5) which is hypothesized to drive the formation of coherent vortex structures (CVS), as Fig. 4b, c shows. CVS presents as continuous and large flow topologies in a frozen fluid domain, which persists longer in time than turbulent eddies in a dissipative scale[32]. The rotation carried out by longitudinal vortex lines periodically lifts the flow away from the boundary wall[33], reducing the intensity of harmonic aerodynamic load in flow-wall interaction (i.e., dipole pressure strength corresponding to tonal noises). As illustrated in Fig. 4d, the production of CVS serves to suppress dipole pressures at 2000

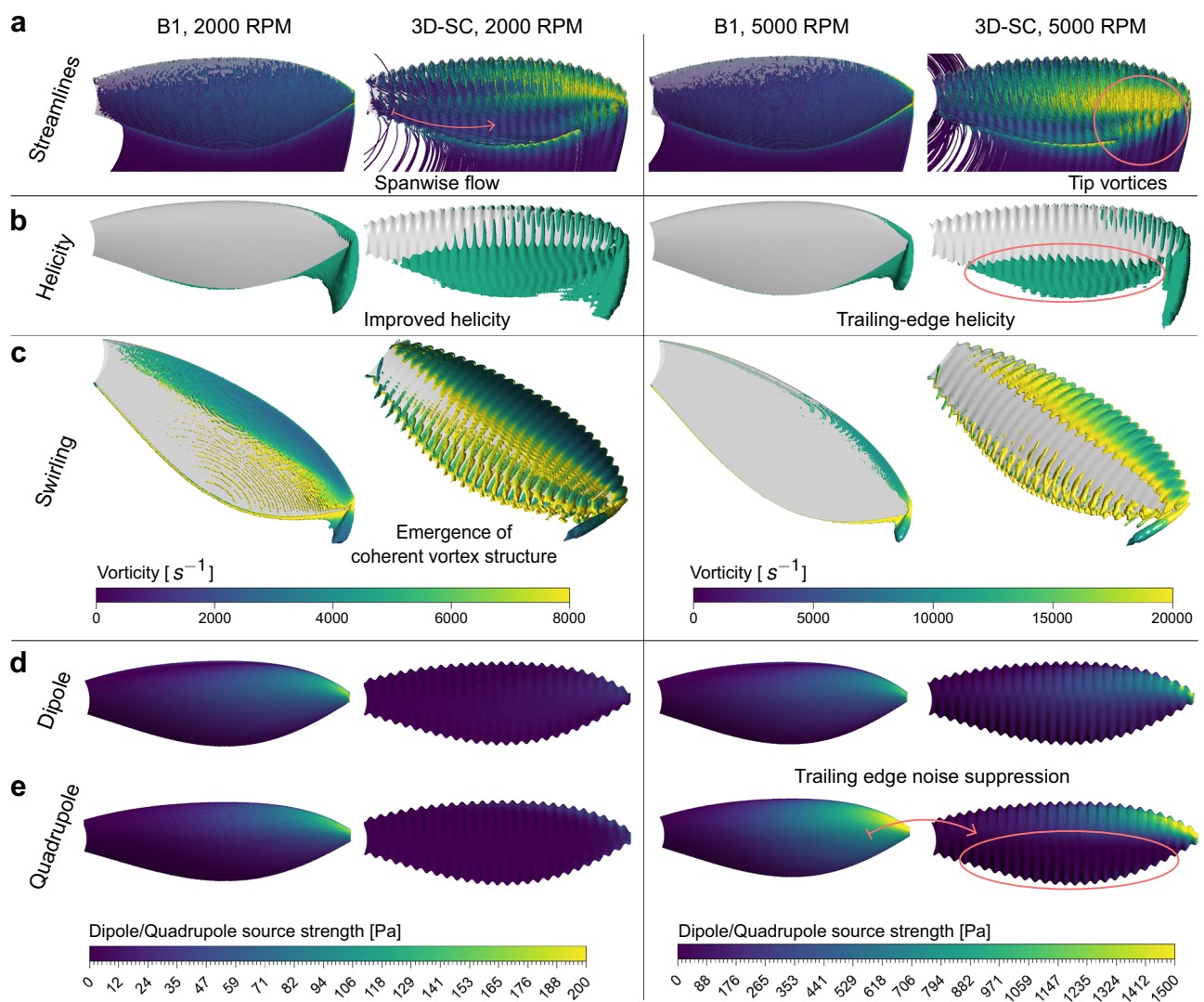

**Fig. 4 | CFD simulation comparisons of B1 and 3D-SC propeller topologies at 2000 and 5000 RPM. a** Streamlines colored by local flow vorticity, illustrating flow pattern variations over the B1 and 3D-SC propellers' surfaces. **b** Helicity iso-surfaces at threshold values of $1.6 \times 10^4 m/s^2$ at 2000 RPM and $1.4 \times 10^5 m/s^2$ at 5000 RPM. **c** Swirling strength iso-surfaces generated at thresholds of $1.4 \times 10^3 s^{-1}$ at 2000 RPM and $5.0 \times 10^3 s^{-1}$ at 5000 RPM, color coded by vorticity with magnitudes represented in the accompanying color scale. **d** Surface dipole pressure source strength contour indicative of tonal noise distribution. **e** Surface quadrupole pressure source strength contour representing the distribution of broadband noise levels.

RPM. This explains why the peak SPLs of multiple acoustic tones of the 3D-SC propeller are measured to be lower than any other benchmark propeller.

At an increased Re number of 1.9e4 at 5000 RPM, inertial forces become dominant over viscous forces, turning any small perturbations into turbulence, as indicated by the turbulence kinetic energy contour shown in Supplementary Fig. 6. This transition is evident in the sound spectrum, as tonal noises gradually become less pronounced amidst broadband noises, accompanied by a surge in the average SPL value. By comparing the helicity and swirling iso-surfaces of the 3D-SC propellers at 2000 and 5000 RPM in Fig. 4b, c, we notice that CVS cannot withstand excessive disruption and decompose as Re increases. Despite the overall abatement, the plot illustrates that the 3D-SC propeller maintains a more expansive helicity region compared to the B1 prototype at 5000 RPM, especially in the vicinity of the propeller's trailing edge. By comparing the dipole and quadrupole pressure contours of the 3D-SC propeller at 2000 and 5000 RPMs in Fig. 4d, e, it is noteworthy that the quadrupole (broadband) pressure sources come into dominance at 5000 RPM. In this context, the 3D-SC propeller correlates to significant suppression of quadrupole source strength near the blade's trailing

edge, as shown in Fig. 4e, which coincides with the CVS-dense region shown in the helicity and swirling strength iso-surfaces. This observation suggests that the production of CVS induced by 3D surface serrations contributes to broadband noise reduction at high rotational speeds, consistent with our experimental data.

To this end, our computational results show that 3D surface serrations can passively reduce noises at various rotational speeds because of the unique pressure distribution pattern created by the valley-like surface topography. The resultant pressure distribution encourages the formation of CVS across the boundary layer that tends to weaken the loading noise associated with laminar flow dynamics at low Re. In flows with high Re, the mechanism effectively attenuates broadband noise by preserving the flow inertia that stabilizes the directional momentum. This stabilization impedes the cascade of energy from large-scale CVS to smaller-scale turbulent eddies. These turbulent eddies are a principal source of broadband noise, owing to their erratic and high-frequency motion patterns. The mechanism thus prolongs the lifespan of CVS, mitigating the premature transition to a fully turbulent state characterized by stochastic broadband noise generation.

## Discussion

In summary, this work on the 3D-SC propeller topology, inspired by the morphological traits of cicadas and owls, has demonstrated a reduction in propeller noise across the frequency spectrum alongside improvements in aerodynamic efficiency compared to benchmark designs. The implementation of 3D serrations yields more substantial noise reduction beyond that of 2D counterparts due to the extended surface textures. In addition, comparative analyses with benchmark designs have demonstrated that incorporating a cicada wing planform notably augments thrust and propulsive efficiency. Despite the intrinsic difference in the operating Re characterizing the flight of owls and cicadas, the amalgamation of these two distinct morphologies leads to a concurrent enhancement of aerodynamic efficiency and noise suppression. We would like to highlight that such improvements are unattainable through either geometric feature in isolation. They are uniquely a result of the synergistic integration of these morphological elements. Furthermore, our parametric study has highlighted the sensitivity of the 3D-SC propeller's performance to the geometric parameters of the serrations, particularly the amplitude and wavelength of the sinusoidal waveform. This underscores the potential for design tuning to achieve precise noise attenuation targets. Computational fluid dynamics simulations have shed light on the mechanisms underpinning these advancements. In particular, the 3D serrations on the propeller blades are effective in generating coherent vortex structures, which play a crucial role in noise suppression. At low Re, these structures help to reduce harmonic aerodynamic loading, weakening the dipole pressure sources that are linked to tonal noises. When operating at higher Re, the enhanced surface circulation from the 3D serrations impedes the breakdown of CVS into eddies of dissipative scales, which in turn lowers the strength of quadrupole pressure sources associated with broadband noises. This intricate interplay between the geometric parameters of the 3D-SC topology and their impact on vortex dynamics and noise attenuation provides a promising avenue for the future design of aerial vehicles that can selectively mitigate noise while maintaining or even enhancing, aerodynamic efficiency.

## Methods

### Acoustic testing platform

The acoustic testing platform consists of two primary systems: the propeller propulsion system and the sound data acquisition system. The propulsion system comprises several integral components. An HRB 6000 mAh 3 S Lithium Polymer (LiPo) Battery is utilized to supply power. A Racerstar Motor Thrust Stand V3 is employed, which incorporates a high-fidelity thrust sensor. In addition, a 30 Ampere Brushless Motor Electric Speed Controller is utilized to regulate the power supply to the Readytosky 2212 920KV Brushless Motor. For experimental trials, the thrust stand and battery are positioned on a tripod to facilitate stability and precision during data collection. The rotational speed of the motor is measured using a NEIKO 20713 A Digital Tachometer, which reports in revolutions per minute (RPM). The sound data acquisition system consists of a miniDSP UMIK-1 omnidirectional USB microphone to measure the acoustic data, which is mounted on a microphone stand during experiments (as shown in Supplementary Fig. 7). We use the REW Room EQ Wizard (REW) as an acoustic measurement software. The SPL data is obtained at a sampling rate of 48 kHz with no timing difference in the sweep. Three sound measurements are taken for each data point and the average of these measurements is taken as the reference value.

### Aerodynamic testing platform

In our aerodynamic testing procedures, we rely on the TYTO Series 1585 thrust stands, as depicted in Supplementary Fig. 8. This thrust stand offers real-time measurements for crucial parameters such as electrical power input, torque, thrust, and rotational speed, consistently sampled at 40 Hz. Each data point plotted in the thrust and mechanical power curves within the manuscript represents the average of 100 individual measurements. For precise RPM measurements, we utilize a TYTO optical RPM probe instead of the built-in motor RPM sensor. The thrust stand does not facilitate direct control of thrust; rather, adjustments to the ESC throttle result in increments of rotational speed and thrust. This control pattern yields a nonlinear adjustment of thrust and torque over time. Nonetheless, the continuous increment of the ESC throttle enables the extraction of smooth thrust and torque curves in relation to the rotational speed. To quantify the measurement uncertainty, we collected 1000 data points at each thrust level and the statistical distributions of these measurements are detailed in Supplementary Fig. 3.

### Fabrication

We utilize the Polyjet 3D printer from Stratasys to fabricate all prototypes tested in this study. The printing resolution is approximately 32 microns in layer thickness. In terms of the material, we select digital ABS because of its high stiffness. The post-processing procedure commences with wet sanding to smooth the surface, followed by an epoxy coating to reinforce the structure, and concludes with a spray paint application to reduce surface friction. The application of epoxy and spray paint poses a risk of altering the designed 3D sinusoidal serrations due to the potential accumulation of material at the edges. To counter this, we maintained a minimal layer thickness during application and employed various drying orientations to prevent the material from pooling. This post-processing sequence yields propellers with high surface smoothness, negligible voids, and enhanced structural strength.

### Computational simulations

Our CFD model consists of mesh generation, ANSYS CFX configuration, and ANSYS CFX post-processing. CFX is an integrated module of Ansys Workbench that allows users to simulate the flow field of interest. In the fluid domain setup, cylindrical rotational and stationary fluid regions are constructed. In general, the rotational region represents the section of the fluid domain where the propeller rotates, and the stationary region is part of the fluid domain where the air is initialized to be static to simulate the neighboring fluid domain of the rotor. The rotational region, which encloses the propeller, is defined by a buffer zone with a height of 10 mm both front and back. In total, -10 million tetrahedron elements are used to construct the rotational domain and most of the cells are allocated to the inflation layers adjacent to the propeller surface. The static region enclosing the rotational region has a radius of 450 mm and a distance of 500 mm to the inlet and 600 mm to the outlet. In total, approximately 2 million tetrahedron elements are used to construct the static region (see more details in Supplementary Fig. 9). The size of the domain is chosen via iterative simulations until numerical convergence is achieved in both the freestream and wake flow. The reference pressure of both regions is set to be 1 atm to keep in accordance with lab conditions. In terms of the boundary conditions, the fluid domain consists of a pressure inlet and outlet of 0 atm gauge pressure, interfaces connecting the rotational and stationary fluid regions, a non-slip propeller surface wall, and a cylindrical slip wall (i.e., zero shear stress) enclosing the static fluid region. The configuration for these connection interfaces employs a 'frozen rotor' setting with the pitch angle set to 360 degrees.

## Data availability

The source data generated in this study are available at https://doi.org/10.5281/zenodo.11088631.

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

## Acknowledgements

This research was supported by CITRIS and the Banatao Institute, Air Force Office of Scientific Research (Fund number: FA9550-22-1-0420), and National Science Foundation XSEDE Supercomputing Resources (Fund Number: ACI-1548562).

## Author contributions

G.G. supervised and designed the research project. Z.W. and S.W. developed the initial design concepts and CAD models. S.F. created the CAD models for parametric study. N.C., S.S., S.F., S.W., and Z.W. fabricated the propeller prototypes. Z.W., S.W., S.F., S.S., N.W. conducted the acoustic and aerodynamic experiments. Z.W. and S.S. conducted the computational fluid dynamics simulations. Z.W., S.W., M.H., K.D., G.G. and all other authors contributed to the data analysis and writing of the manuscript.

## Competing interests

Z.W., S.W., S.F., N.C., N.W., S.S., and G.G. have submitted a patent disclosure related to this paper. All other authors declare no competing interests.
