## [Peer Review File · Nature Communications]

Towards silent and efficient flight by combining bioinspired owl feather serrations with cicada wing geometryREVIEWER COMMENTS

Reviewer #1 (Remarks to the Author):

In this paper, the authors develop a propeller blade that incorporates serrations along the surface (inspired by the leading-edge serrations of owls) and a planform shape (inspired by cicada wings). Experimental and numerical testing is performed. Experimental results indicate the new blade outperforms a traditional propeller both acoustically and in aerodynamic performance. The paper is well written, and the figures are nicely made. Including both experimental and numerical results is a great idea however the outputs from these two methods should also be used to provide validation rather than to only provide additional data. I have some concerns relating to the implementation of the comparisons at different stages of the work.

Major items:

Why does B2 not have both trailing and leading-edge serrations as the 3D-SC does? Previous studies have found a significant role of both edge serrations on aeroacoustics. It seems this would mask some implications and challenge the interpretation of your results. Please discuss why this is a fair benchmark.

It appears that B3 is not better for either aerodynamics or sound reduction, not that it sacrificed one for the other. How did you choose the B3 blade shape, was this an engineered optimized shape? Is this a truly fair comparison to the existing field?

The supplemental material mentions a comparison between SST and LES models, what were those outputs? How did they agree with (or not agree with) your experimental results? Please provide a plot of aerodynamic performance as measured experimentally and computationally to give confidence that these methods agree and when/where they don't agree.

Minor items:

Page 2:

Highlighting the large differences in Reynolds numbers between owls and insects should be discussed in the introduction. Especially in aeroacoustics where shedding patterns governs a lot of noise generation. Why would we expect that combining these features, which come from very different uses in nature, on an engineered rotating wing to improve performance? Discuss the equivalent conditions the props are tested at here to help the reader a comparison.

Re-word: "for quietness" to something such as "that has been linked to noise reduction". Proving a biological feature exists for one specific function is challenging and best to avoid that implication.

It is a stretch to call the wavy surface of the blade "owl-inspired". I would add one extra sentence in the intro to make it clear that this implementation is unlike owl wings but is inspired by one specific feature extended across the whole surface.

Page 6:

I do not believe it is true that surface friction drag is ruled out because the surface material and area is the same. Serrations have been introduced across the surface of these wings that will create a sinusoidal no-slip boundary condition, thus directly contributing to vorticity generation and drag. This is independent from turbulence. The drag related paragraph should be revised.

Reviewer #2 (Remarks to the Author):

In this manuscript, the aerodynamic and aeroacoustic performance of propellers with a three-dimensional surface structure is investigated. The authors investigate the performance experimentally, and then numerically investigate the mechanisms by which the differences are caused. The acoustic performance of propellers, which is the subject of this paper, is of great importance for the expansion of drone applications. However, there are several serious concerns and I cannot recommend the acceptance for publication.

1. I understand that the propeller noise is measured by a microphone located on the axis of rotation of the propeller in this study. Propeller noise is strongly dependent on the relative location and distance of the microphone. On the axis of rotation, noise at the blade-pass frequency such as dipole-type sounds (Gutin sound) induced by the aerodynamic forces are minimised due to the small changes in the orientation of the propeller's surface with respect to the microphone. On the other hand, the noise we hear when drones fly is dominated by the Gutin sound at the blade-pass frequency, as we do not hear the noise on the axis of rotation. The comparison in this study is made without any consideration of this dominant sound, and it is necessary to show that the surface structure is effective in reducing the sound at different directions and distances.

2. Assuming the above experiment has been carried out, comparisons need to be made fairly. The propeller is the element used to generate thrust and the noise needs to be evaluated at the same thrust, not at the same rotation speed as shown in figure 2b,c.

For example, according to previous studies, the noise levels of dipole type and quadropole type sound sources are proportional to the sixth and eighth power of the rotation speed, respectively (Ffowcs Williams JE, Hawkings DL. 1969 Theory relating to the noise of rotating machinery. J. Sound Vib. 10, 10–21). In figure 2d, there are differences in the thrust between designs at the same rotation speed. For other propellers to generate the same thrust as the B2, they would need to increase their rotational speed, which would increase the noise level significantly. The B2 design has a higher thrust coefficient, which means that the rotational speed can be suppressed and the noise level for the same thrust may be reduced with B2 design, but the improvement in efficiency looks marginal. In any case, there are strong doubts as to whether the noise level of 3D-SC would really be reduced when equipped with a flying drone.

Reviewer #3 (Remarks to the Author):

+++++ SUMMARY +++++

The authors have conducted a study with the objective of enhancing the performance of traditional propeller systems. Specifically, they have investigated the combined advantages of incorporating features from owl wings and cicada wings. Owl wing geometry has been leveraged to reduce noise, while cicada wing characteristics have been introduced to mitigate aerodynamic performance degradation. This research encompasses both experimental and computational approaches. The manuscript discusses the benefits of the proposed solution, focusing on improvements in aerodynamic efficiency and noise reduction, and supported by a selection of findings. Overall, the manuscript is well-structured and written in a clear manner.

+++++ MAIN CRITICISMS +++++

This reviewer encounters difficulty in connecting the present work with the current state of the art. Is this the first time in which the advantages of bio-inspired planforms have been combined with bio-inspired trailing edge serrations? What types of planforms have previously been employed in conjunction with serrated trailing edge (TE) propellers?

In the opinion of this reviewer, the followed methodology does not allow to directly conclude on the benefits of the proposed technology, and the work could be considered a preliminary study. It remains unclear what the optimal operating parameters are for each of the proposed designs, and

e.g. no consideration was given to pitch variation. Additionally, the intended application for this technology is not clear to this reviewer. Factors such as scale effects, manufacturability, and durability would play a pivotal role when translating the findings of this study into industrial applications.

There seems to be no information on how the authors dealt with the uncertainties of their experiments.

+++++ OTHER COMMENTS +++++

[Abstract and Introduction] The translation of bio-inspired shapes into the geometrical adaptation of the propeller could be made clearer. Specifically, the utilization of owl wings as inspiration for defining TE serrations and cicada wings for the planform should be explicitly explained.

[Page 1] This reviewer has difficulty understanding the choice of using B-splines as a constraint for design. In the present study, the authors employed trigonometric functions and polynomials to represent their geometry, which are arguably more restrictive than B-splines.

[Page 3] How did the authors design the serrations and the cicada planform? Was it based on the study and scaling of a single individual, or did it result from statistical analysis?

[Page 4] Why did the authors consider the effective angle of attack in Equation 6? This reviewer is not familiar with such an approach. Does this definition align with the information provided in the supplementary material?

To enhance comprehension of this work, certain essential information found in the supplementary material could be integrated into the main manuscript.

+++++ FORMAL ASPECTS +++++

On page 10, the "Conclusions" should be given a dedicated section, as they currently appear as a paragraph within the results section.

Response to reviewers' comments

We thank the reviewers for their insightful comments and critique of the presented work. Please find below a point-by-point response to every comment and indications of what and where changes have been made. These changes are indicated in blue text in the revised paper.

Reviewer #1 (Remarks to the Author):

In this paper, the authors develop a propeller blade that incorporates serrations along the surface (inspired by the leading-edge serrations of owls) and a planform shape (inspired by cicada wings). Experimental and numerical testing is performed. Experimental results indicate the new blade outperforms a traditional propeller both acoustically and in aerodynamic performance. The paper is well written, and the figures are nicely made. Including both experimental and numerical results is a great idea however the outputs from these two methods should also be used to provide validation rather than to only provide additional data. I have some concerns relating to the implementation of the comparisons at different stages of the work.

Major items:

Why does B2 not have both trailing and leading-edge serrations as the 3D-SC does? Previous studies have found a significant role of both edge serrations on aeroacoustics. It seems this would mask some implications and challenge the interpretation of your results. Please discuss why this is a fair benchmark.

We thank the reviewer for bringing this to our attention. We agree with the observation that the B2 prototype can include both edge serrations in the design. In response to this comment, we have included a redesign of the B2 prototype with both edge serrations. The updated discussions and results are shown in our detailed response to the following comment below.

It appears that B3 is not better for either aerodynamics or sound reduction, not that it sacrificed one for the other. How did you choose the B3 blade shape, was this an engineered optimized shape? Is this a truly fair comparison to the existing field?

We extend our gratitude to the reviewer for this valuable comment. The choice of the B3 prototype is intended to showcase the characteristics of a conventional propeller planform. This planform distinctly positions the maximum chord at approximately 20-25% of the spanwise direction. This design paradigm is derived from precedents within the field^{1,2}, as it has been utilized in prior studies that explore the impact of serrations on the aerodynamic performance of conventional propeller designs. The B3 prototype is thus indicative of the most ubiquitous yet non-optimized aerodynamic planform observed in standard propeller design.

We acknowledge the reviewer's comment regarding the benchmarking of our propeller designs against industry standards. To address this, we have introduced a B4 prototype as a benchmark in our study to provide a comparison with a state-of-the-art propeller manufactured by DJI Technology Co., Ltd. This addition offers a pertinent industry reference point, showcasing where our design stands relative to current technological advancements. In response to the feedback from another reviewer, we have also upgraded our aerodynamic testing platform to facilitate a more accurate and reliable comparison of thrust and aerodynamic efficiency. Our updated results indeed reveal that the B3 prototype is outperformed by the B4 model in both aerodynamic efficiency and acoustic signature. However, it is critical to emphasize that the 3D-SC propeller design, which is the main focus of our research, demonstrates superior performance over the B4 prototype. These findings are pivotal as they underline the potential of the 3D-SC propeller in advancing propeller technology beyond the current capabilities offered by existing models. The updated experimental results and corresponding discussions are presented as follows:

Updates in section **Introduction**:

“The comparison between B1 and B3 provides insights into the impact of the cicada planform. Furthermore, we introduce the DJI Phantom 3 propeller as an additional benchmark (B4), which serves as a reference for current state-of-the-art industry standards.”

Updates in section **Acoustic advancement at various rotational speeds in experiments** updates:

“Acoustic measurements of the 3D-SC and other benchmark prototypes are conducted at two distinct radial distances - 0.1 meters and 5 meters from the rotor - providing a near-field measurement and a relatively more distant comparative measurement. At each radial position, the microphone is circumferentially positioned around the thrust stand to collect sound data from different angles. In the experiment, the sound spectra of each prototype, corresponding to different thrusts and radial distances, are recorded from 0 to 2 kHz. The sound spectrum is subsequently integrated to evaluate OASPL, as shown in **Fig. 2.b**. These measurements are taken with the microphone positioned directly in front of the rotor (0°). The trend indicates that the OASPL of rotor noise generally increases with thrust and decreases with the radial distance. Experimental results show the cicada wing inspired B1 prototype achieves up to 1.6 dB lower OASPL than the conventional B3 prototype at 0.1 meters with 50 gram-forces (gf) thrust, reducing to 1.4 dB at 5 meters. By contrast, the implementation of 3D serrations carries out a more significant reduction in noise. The 3D-SC model showcases a noise decrease of 8.5 dB relative to the B1 model under identical conditions (0.1 m, 50 gf). This effect of 3D serrations is consistently observed in its modification to the conventional planform. In particular, the OASPL of B2 is 8.8 dB lower than that of B3. In the context of industry benchmarks, the OASPL of the B4 prototype is reduced in comparison to the non-serrated models (B1 and B3), yet it remains higher than that of the serrated designs (3D-SC and B2). Notably, the 3D-SC prototype demonstrates a reduction of 5.0 dB when benchmarked against B4 at 5 m, 50 gf.

Fig. 2c presents the sound distribution in a polar coordinate system at distances of 0.1 m and 5 m, with radial and angular coordinates representing the OASPL and the measurement angle, respectively. A constant thrust of 50 gf is applied in this evaluation to simulate a high-thrust scenario akin to that of a drone propeller during flight. The level of noise reduction is more pronounced at the 0.1 m measurements. As the measurement distance increases, the amount of reduction declines accordingly. For a quantitative assessment of acoustic performance, a measurement distance of 5 meters is selected to emphasize drone noise as perceived in the far-field. The peak noise abatement is achieved at a 30° measurement angle, where the 3D-SC design demonstrates a 5.5 dB noise reduction compared to the industry benchmark B4. The analyses herein utilize the unweighted OASPL. However, for industry relevance, the A-weighted OASPL—which accentuates frequencies within the human auditory range—is often preferred. When examining frequencies critical to human hearing, the 3D-SC prototype delivers a maximum noise reduction of 3.3 dBA compared to the B4 design at a 30° measurement angle (see Supplementary Information **Fig. S.3a-b**).

To explore the effective noise attenuation bandwidth, an analysis of the rotor sound spectrum is conducted. Sound spectra at 15 gf (corresponding to $Re \approx 1.01e4$) and 50 gf (corresponding to $Re \approx 1.82e4$) are chosen to exemplify the acoustic characteristics of the propellers across different flow regimes, as depicted in **Fig. 2d**. The spectral analysis employs a 5th-order Savitzky-Golay filter²⁵, which was employed to smooth the data and reduce noise without significantly distorting the signal. The rotor spectrum at 15 gf is dominated by acoustic tones whose frequencies are lower than 10 kHz. This confirms that loading noise and its harmonics are the primary component of propeller noise at low rotational speeds. Notably, the SPLs of multiple tones of a 3D-SC propeller are apparently lower than B1-4. At higher Reynolds numbers represented by 50 gf, these SPL peaks diminish in prominence, giving way to broadband noise as the primary acoustic signature. A comparative study of the B1 and B3 prototypes reveals that the cicada wing-inspired planform alone has a limited effect on noise mitigation, increasing peak SPL within the 5-9 kHz frequency band while reducing noise elsewhere. In contrast, the

implementation of 3D serrations results in a SPL reduction across the entire spectrum. These experimental results of rotor noise across various thrusts and locations finds that the 3D-SC design greatly reduces noise levels compared to other benchmarks.”

Updates in section **Experimental evidence of 3D-SC aerodynamic advantages:**

“In terms of aerodynamic efficiency, the mechanical power, $\mathcal{W} = Q \cdot n$, of a propeller is a critical determinant of energy consumption rates during flight, where Q is the propeller torque.

As depicted in **Fig. 2e**, the thrust generation of propellers is assessed across the rotational speeds up to 6000 RPM, with average mechanical power measured at thrust levels from 0 to 50 gf. The 3D-SC design outperforms other models across all speeds, notably at higher RPMs. A notable thrust enhancement of 14.8 gf, or 39.2%, is observed in the B1 versus B3 comparison at 5000 RPM, credited to the aerodynamic refinement inherent in the cicada wing planform. At the same RPM, the 3D-SC design produces an extra 20.3 gf of thrust over B4. Analysis of the thrust coefficient reveals that the 3D-SC prototype exhibits a coefficient that is 0.04 higher than that of the B4 model, representing a substantial enhancement of 55.6% (see Supplementary Information **Fig. S.3c**).

Efficiency assessments suggest that the 3D-SC design not only boosts thrust but also diminishes power consumption, thus enhancing propulsive efficiency. When assessed at an equivalent thrust of 50 gf, the 3D-SC prototype exhibits a reduction in mechanical power consumption by 0.17 W, an improvement of 4.1% compared to the B1 design, indicating the marginal efficiency benefits conferred by 3D serrations. In contrast, the B1 versus B3 comparison highlights a significant efficiency leap with the cicada wing planform, achieving a 1.49 W power reduction, equating to a 26.7% decrease. Against the B4 design, the 3D-SC model exhibits a substantial 20.2% cut in power consumption, reinforcing its dual advancements in thrust production and energy savings. Confined by the limited strength of digital ABS material, the rotor diameter is set as 6 inches, allowing for the assessment of all prototypes across a wider spectrum of rotational speeds without causing significant elastic deformation. This dimension is notably smaller than the propellers commonly employed in drones³. To investigate the scalability of our findings for larger-scale applications, additional experiments are performed on 12-inch 3D-SC and B4 propeller prototypes, operating at a maximum rotational speed of 3000 RPM. The outcomes of these tests are detailed in **Fig. S.4**, enabling the assessment of propeller performance at elevated thrust levels. At an equivalent thrust of 150 gf, the 3D-SC propeller achieves a mechanical power reduction of 2.29 W, corresponding to an improvement of 22.6% over the B4 prototype.”

Updates in **Supplementary Information:**

“**Aerodynamic Testing Platform:** We utilize the TYTO Series 1585 thrust stand for all aerodynamic tests, as shown in **Figure S8**. This device delivers real-time measurements of various parameters, including the rotor’s electrical power input, torque, thrust, and rotational speed, with a uniform sampling rate of 40 Hz. To improve the data accuracy, each thrust and mechanical power curve presented in the manuscript comprises of 100 data points. Mitigating measurement uncertainty further, each data point is taken by the average of 100 measurements. Notably, the measurement steadiness decreases as the rotational speed increases. Nevertheless, the maximum level of uncertainty is maintained within an acceptable range. Specifically, at 50 gf thrust, the coefficient of variation for thrust and torque measurements is recorded at 0.33% and 2.2% respectively. Owing to the employment of a TYTO optical RPM probe, the measurement error in rotational speed is negligible.”

Updated **Figures:**

Fig. 1. Illustration of the design concepts for a 3D-SC propeller. (a) 3D-SC topology inspired by owl feather and cicada forewing geometry. (b) Fabricated 3D-SC, B1, B2, B3, and B4 prototypes using Polyjet additive manufacturing

Fig. 2. Aerodynamic and aeroacoustic experimental results. (a) CAD schematics of 3D-SC, B1, B2, B3, and B4 propeller designs with associated color codes. (b) OASPL data across thrusts ranging from 10 to 50 gf. (c) Sound spectra comparison at 0.1 m (top) and 5 m (bottom) radial distances under 50 gf thrust conditions. (d) Polar representation of OASPLs at different measurement angles, with data collected at radial distances of 0.1 m (left) and 5 m (right). (e) Graphs depicting the relationship between thrust and rotational speed (upper) and mechanical power versus thrust (lower).

Fig. S.3: Supplementary experimental results. (a) A-weighted OASPL across different thrust levels. (b) Polar representation of A-weighted OASPL at varying measurement angles. (c) Thrust coefficient against rotational speed.

Fig. S.4, Comparative aerodynamic analysis of 12-inch 3D-SC and B4 prototype models. (a) The 3D-printed prototypes featuring the 12-inch model above and the 6-inch model below. (b) Thrust versus rotational speed plot (left) and mechanical power against thrust plot (right).

Fig. S.8: Experimental setup for aerodynamic data collection.

The supplemental material mentions a comparison between SST and LES models, what were those outputs? How did they agree with (or not agree with) your experimental results? Please provide a plot of aerodynamic performance as measured experimentally and computationally to give confidence that these methods agree and when/where they don't agree.

We appreciate the reviewer's feedback. Upon detailed examination of the CFD results obtained from the Shear Stress Transport (SST) and Large Eddy Simulation (LES) solvers, it became apparent that the SST model exhibits a larger error in its predictions. While both solvers provide precise thrust estimates, the LES model proves to be superior in accurately predicting rotor torque. In response, we have included discussions in the Supplementary Information that detail the differences between SST and LES simulations, as well as how they correlate with our experimental results:

Updates in **Supplementary Information:**

“Aerodynamic modelling

For enhanced precision in computational fluid dynamics (CFD) analyses, the selection of an appropriate turbulence model is essential. Our methodology incorporates the Large Eddy Simulation (LES) approach for simulating the turbulent flow characteristics, in contrast to the Menter Shear-Stress Transport (SST) model. The SST model integrates blending functions to segregate the far-field and near-wall domains, while it employs the Reynolds Averaged Navier-Stokes (RANS) framework for the calculation of averaged flow properties with additional fluctuating components. In distinction to RANS, the LES methodology is capable of resolving larger scale vortical structures via sophisticated filtering techniques. The sub-grid scale (SGS) model inherent in LES excludes the smaller scales of motion that have a minimal impact on energy, thereby streamlining computational resources toward scales that are more significant dynamically. LES specifically concentrates on the scales of energy transfer, which are crucial for accurately predicting the flow's physical behavior. Within LES, the formulation of momentum balance is carefully designed to encapsulate these critical factors, as shown below.

$$\rho \left(\frac{\partial \hat{u}_i}{\partial t} + \frac{\partial \hat{u}_i}{\partial x_j} \hat{u}_j \right) = - \frac{\partial p}{\partial x_i} - \lambda \frac{\partial}{\partial x_i} \left(\frac{\partial \hat{u}_k}{\partial x_k} \right) + \frac{\partial}{\partial x_j} \left(\tau_{ij}^{les} + \tau_{ij}^{sgs} \right) \quad (3)$$

For scales larger than the grid filter, the velocity field undergoes spatial filtering via a convolutional integral, which is mathematically represented by the subsequent equation:

$$\hat{\mathbf{u}}(\mathbf{x}, t) = \int_{-\infty}^{\infty} G(\mathbf{x} - \boldsymbol{\xi}) \mathbf{u}(\boldsymbol{\xi}, t) d\boldsymbol{\xi} \quad (4)$$

where \mathbf{x} represents the coordinates of the point where the filter is applied, $\boldsymbol{\xi}$ is the variable of spatial integration, and $G(\mathbf{x} - \boldsymbol{\xi})$ is the Gaussian spatial filter. As eddy sizes reduce, so does the eddy viscosity within the LES framework. The Smagorinsky model¹ is utilized to calculate eddy viscosity across various turbulent scales, as demonstrated in the following expressions:

$$\tau_{ij}^{sgs} = - \frac{\partial}{\partial x_j} \rho \widehat{u'_i u'_j} = \mu_t \left[\frac{\partial \hat{u}_i}{\partial x_j} + \frac{\partial \hat{u}_j}{\partial x_i} \right] \quad (5)$$

$$\mu_t = \rho (C_s \Delta)^2 \cdot \hat{S} \quad (6)$$

In this context, u'_i and u'_j represents the velocity profile of sub-grid eddies, μ_t denotes the corresponding dynamic viscosity, Δ represents the sub-grid length scale, C_s represents the Smagorinsky constant, and \hat{S} stands for the resolved strain rate. In this sense, u'_i is not resolved in the solver. Instead, the complex interactions of these sub-grid eddies, typically expressed by the term $\frac{\partial}{\partial x_j} \rho \widehat{u'_i u'_j}$ is approximated by a viscous stress tensor τ_{ij}^{sgs} to constitutively compensate the energy loss due to the cut-off of u'_i term (i.e., spatial filtering).

The comparison between LES and SST computational models against experimental benchmarks, depicted in **Figure S10**, indicates a minor discrepancy in thrust predictions, peaking at 0.44 gf (0.8 %) at 5000 RPM. Torque predictions, however, diverge more significantly, with errors increasing alongside rotational speeds. At 5000 RPM, the SST model's highest error registers at 0.0011 Nm (12.4 %), contrasted by the LES model's more modest 0.00049 Nm (5.4 %) discrepancy. Overall, LES provides a closer approximation of the propeller's aerodynamic performance.

Fig. S.10: Validation of different CFD turbulence models against experimental data for (a) thrust and (b) torque as functions of rotational speed.”

Minor items:

Page 2:

Highlighting the large differences in Reynolds numbers between owls and insects should be discussed in the introduction. Especially in aeroacoustics where shedding patterns governs a lot of noise generation. Why would we expect that combining these features, which come from very different uses in nature, on an engineered rotating wing to improve performance? Discuss the equivalent conditions the props are tested at here to help the reader a comparison.

We express our gratitude to the reviewer for the valuable comment. Indeed, the flight dynamics of cicadas and owls are characterized by distinct Reynolds numbers. Our decision to integrate these two features stems from empirical evidence demonstrating a general aerodynamic penalty that's induced by the implementation of owl-inspired 2D serrations. This concept is discussed in the introduction:

“However, in conventional designs such as the leading-edge sawtooth serration, slitted serration, and sinusoidal serration, the pursuit of passive noise reduction comes at a penalty of overall aerodynamic performance.

In this work, we formulate new design strategies that can mitigate tradeoffs between noise reduction and aerodynamic performance by merging owl feather and cicada insect wing geometries to create a three-dimensional topology that features silent and efficient flight.”

The propeller design inspired by the cicada wing has been demonstrated to significantly improve aerodynamic performance, an aspect discussed in the introduction:

“Specifically, insect wing geometries such as the cicada have been established as promising templates in the design of aerodynamically advanced devices and propellers. Therefore, we have adopted the wing shape of the cicada as a synergistic planform, integrated to address the demands of advanced aerodynamic performance.”

In this sense, we anticipate that combining these bio-inspired features will mitigate the limitations of each and yield a propeller design that excels in both aerodynamics and aeroacoustics.

In response to the reviewer's remarks regarding Reynolds number considerations, it is acknowledged that the flight of owls intrinsically differ from that of cicadas in terms of Reynolds numbers. Specifically, the Reynolds number for cicada flight is estimated at 1983^3 , while owl flight typically exhibits a Reynolds number around $1.5e5^4$. Despite these differences, existing research suggests that cicada wing planforms offer aerodynamic benefits across a wide range of Reynolds numbers, including those at which conventional drone propellers operate⁵. Therefore, Reynolds number was not a primary driver in the conceptualization of our design.

Re-word: “for quietness” to something such as “that has been linked to noise reduction”. Proving a biological feature exists for one specific function is challenging and best to avoid that implication.

We appreciate this valuable suggestion. We have refined our statements accordingly:

“Researchers are alternatively exploring the flight mechanisms of natural predators renowned for their stealth, aiming to derive innovative and efficacious design principles.”

It is a stretch to call the wavy surface of the blade “owl-inspired”. I would add one extra sentence in the intro to make it clear that this implementation is unlike owl wings but is inspired by one specific feature extended across the whole surface.

We appreciate the reviewer's comment and have revised the relevant statement in the **Introduction** accordingly:

“Specifically, our design introduces a high-fidelity, three-dimensional (3D) sinusoidal serration topography that encompasses a widespread surface adaptation rather than a localized edge variation for potential acoustic improvement, as illustrated in **Fig. 1a**. Integrating this design, the cicada-inspired wing shape serves as the foundational planform for the propeller to augment aerodynamic efficiency.”

Page 6:

I do not believe it is true that surface friction drag is ruled out because the surface material and area is the same. Serrations have been introduced across the surface of these wings that will create a sinusoidal no-slip boundary condition, thus directly contributing to vorticity generation and drag. This is independent from turbulence. The drag related paragraph should be revised.

We thank the reviewer for this valuable comment. We agree that the surface friction drag is not entirely ruled out given the alignment of surface material and area. We have removed corresponding statement in section **Experimental evidence of 3D-SC aerodynamic advantages**.

Reviewer #2 (Remarks to the Author):

In this manuscript, the aerodynamic and aeroacoustic performance of propellers with a three-dimensional surface structure is investigated. The authors investigate the performance experimentally, and then numerically investigate the mechanisms by which the differences are caused. The acoustic performance of propellers, which is the subject of this paper, is of great importance for the expansion of drone applications. However, there are several serious concerns and I cannot recommend the acceptance for publication.

1. I understand that the propeller noise is measured by a microphone located on the axis of rotation of the propeller in this study. Propeller noise is strongly dependent on the relative location and distance of the microphone. On the axis of rotation, noise at the blade-pass frequency such as dipole-type sounds (Gutin sound) induced by the aerodynamic forces are minimised due to the small changes in the orientation of the propeller's surface with respect to the microphone. On the other hand, the noise we hear when drones fly is dominated by the Gutin sound at the blade-pass frequency, as we do not hear the noise on the axis of rotation. The comparison in this study is made without any consideration of this dominant sound, and it is necessary to show that the surface structure is effective in reducing the sound at different directions and distances.

We thank the reviewer for this helpful feedback. As suggested by the reviewer, we have conducted additional experiments specifically designed to examine the impact of measurement direction and distance on the acoustic profile of the rotor. The results of these experiments have provided further insights into the acoustic behavior of the rotor across various distances and directional orientations. These findings have been integrated into our discussion to offer a more complete understanding of the rotor's acoustic characteristics shown in section **Acoustic advancement at various rotational speeds in experiments** and below. To address the comment from other reviewers, we have introduced the B4 prototype as a benchmark in our study to provide a comparison with a state-of-the-art propeller manufactured by DJI Technology Co., Ltd. This addition offers a pertinent industry reference point, showcasing where our design stands relative to current technological advancements. Additionally, we have adjusted the design of our conventional benchmark, B3, and its serrated equivalent, B2, to guarantee that the pitch angle of the cicada-inspired planform precisely matches that of the conventional planform, ensuring a better comparison.

Updates in section **Acoustic advancement at various rotational speeds in experiments**:

“Acoustic measurements of the 3D-SC and other benchmark prototypes are conducted at two distinct radial distances - 0.1 meters and 5 meters from the rotor - providing a near-field measurement and a relatively more distant comparative measurement. At each radial position, the microphone is circumferentially positioned around the thrust stand to collect sound data from different angles. In the experiment, the sound spectra of each prototype, corresponding to different thrusts and radial distances, are recorded from 0 to 2 kHz. The sound spectrum is subsequently integrated to evaluate OASPL, as shown in **Fig. 2.b**. These measurements are taken with the microphone positioned directly in front of the rotor (0°). The trend indicates that the OASPL of rotor noise generally increases with thrust and decreases with the radial distance. Experimental results show the cicada wing inspired B1 prototype achieves up to 1.6 dB lower OASPL than the conventional B3 prototype at 0.1 meters with 50 gram-forces (gf) thrust, reducing to 1.4 dB at 5 meters. By contrast, the implementation of 3D serrations carries out a more significant reduction in noise. The 3D-SC model showcases a noise decrease of 8.5 dB relative to the B1 model under identical conditions (0.1 m, 50 gf). This effect of 3D serrations is consistently observed in its modification to the conventional planform. In particular, the OASPL of B2 is 8.8 dB lower than that of B3. In the context of industry benchmarks, the OASPL of the B4 prototype is reduced in comparison to the non-serrated models (B1 and B3), yet it remains higher than that of the serrated designs (3D-SC and B2). Notably, the 3D-SC prototype demonstrates a reduction of 5.0 dB when benchmarked against B4 at 5 m, 50 gf.

Fig. 2c presents the sound distribution in a polar coordinate system at distances of 0.1 m and 5 m, with radial and angular coordinates representing the OASPL and the measurement angle, respectively. A constant thrust of 50 gf is applied in this evaluation to simulate a high-thrust scenario akin to that of a drone propeller during flight. The

level of noise reduction is more pronounced at the 0.1 m measurements. As the measurement distance increases, the amount of reduction declines accordingly. For a quantitative assessment of acoustic performance, a measurement distance of 5 meters is selected to emphasize drone noise as perceived in the far-field. The peak noise abatement is achieved at a 30° measurement angle, where the 3D-SC design demonstrates a 5.5 dB noise reduction compared to the industry benchmark B4. The analyses herein utilize the unweighted OASPL. However, for industry relevance, the A-weighted OASPL—which accentuates frequencies within the human auditory range—is often preferred. When examining frequencies critical to human hearing, the 3D-SC prototype delivers a maximum noise reduction of 3.3 dBA compared to the B4 design at a 30° measurement angle (see Supplementary Information **Fig. S.3a-b**).

To explore the effective noise attenuation bandwidth, an analysis of the rotor sound spectrum is conducted. Sound spectra at 15 gf (corresponding to $Re \approx 1.01e4$) and 50 gf (corresponding to $Re \approx 1.82e4$) are chosen to exemplify the acoustic characteristics of the propellers across different flow regimes, as depicted in **Fig. 2d**. The spectral analysis employs a 5th-order Savitzky-Golay filter²⁵, which was employed to smooth the data and reduce noise without significantly distorting the signal. The rotor spectrum at 15 gf is dominated by acoustic tones whose frequencies are lower than 10 kHz. This confirms that loading noise and its harmonics are the primary component of propeller noise at low rotational speeds. Notably, the SPLs of multiple tones of a 3D-SC propeller are apparently lower than B1-4. At higher Reynolds numbers represented by 50 gf, these SPL peaks diminish in prominence, giving way to broadband noise as the primary acoustic signature. A comparative study of the B1 and B3 prototypes reveals that the cicada wing-inspired planform alone has a limited effect on noise mitigation, increasing peak SPL within the 5-9 kHz frequency band while reducing noise elsewhere. In contrast, the implementation of 3D serrations results in a SPL reduction across the entire spectrum. These experimental results of rotor noise across various thrusts and locations finds that the 3D-SC design greatly reduces noise levels compared to other benchmarks.”

Updated Figures:

Fig. 3. Aerodynamic and aeroacoustic experimental results. (a) CAD schematics of 3D-SC, B1, B2, B3, and B4 propeller designs with associated color codes. (b) OASPL data across thrusts ranging from 10 to 50 gf. (c) Sound spectra comparison at 0.1 m (top) and 5 m (bottom) radial distances under 50 gf thrust conditions. (d) Polar representation of OASPLs at different measurement angles, with data collected at radial distances of 0.1 m (left) and 5 m (right). (e) Graphs depicting the relationship between thrust and rotational speed (upper) and mechanical power versus thrust (lower).

Fig. S.3: Supplementary experimental results. (a) A-weighted OASPL across different thrust levels. (b) Polar representation of A-weighted OASPL at varying measurement angles. (c) Thrust coefficient against rotational speed.

2. Assuming the above experiment has been carried out, comparisons need to be made fairly. The propeller is the element used to generate thrust and the noise needs to be evaluated at the same thrust, not at the same rotation speed as shown in figure 2b,c. For example, according to previous studies, the noise levels of dipole type and quadropole type sound sources are proportional to the sixth and eighth power of the rotation speed, respectively (Ffowcs Williams JE, Hawkings DL. 1969 Theory relating to the noise of rotating machinery. *J. Sound Vib.* 10, 10–21). In figure 2d, there are differences in the thrust between designs at the same rotation speed. For other propellers to generate the same thrust as the B2, they would need to increase their rotational speed, which would increase the noise level significantly. The B2 design has a higher thrust coefficient, which means that the rotational speed can be suppressed and the noise level for the same thrust may be reduced with B2 design, but the improvement in efficiency looks marginal. In any case, there are strong doubts as to whether the noise level of 3D-SC would really be reduced when equipped with a flying drone.

We thank the reviewer for this valuable comment. In response, we have re-collected the acoustic profiles of all prototypes, ensuring that each is measured under identical thrust conditions to facilitate a direct comparison. Furthermore, to enhance the accuracy and reliability of our comparisons regarding the propeller's propulsive efficiency, we have implemented substantial improvements to our aerodynamic testing apparatus. The improved thrust stand is capable of directly quantifying both the torque exerted by the propeller and its resulting mechanical power output. This capability enables a more precise method for comparing the propulsive efficiencies of different propeller designs. Following the redesign of the B3 and B2 prototypes, the updated conventional planform exhibits a reduced thrust output, which is also accompanied by a concurrent reduction in torque (drag) relative to our previous designs. Except for this difference, all other trends remain the same as before. The updated acoustic results and discussions are presented in our response above. Detailed descriptions of the updated methodology, along with the revised results and in-depth discussions, are delineated below:

Updates in section **Experimental evidence of 3D-SC aerodynamic advantages:**

“In terms of aerodynamic efficiency, the mechanical power, $\mathcal{W} = Q \cdot n$, of a propeller is a critical determinant of energy consumption rates during flight, where Q is the propeller torque.

As depicted in **Fig. 2e**, the thrust generation of propellers is assessed across the rotational speeds up to 6000 RPM, with average mechanical power measured at thrust levels from 0 to 50 gf. The 3D-SC design outperforms other models across all speeds, notably at higher RPMs. A notable thrust enhancement of 14.8 gf, or 39.2%, is observed in the B1 versus B3 comparison at 5000 RPM, credited to the aerodynamic refinement inherent in the cicada wing planform. At the same RPM, the 3D-SC design produces an extra 20.3 gf of thrust over B4. Analysis of the thrust coefficient reveals that the 3D-SC prototype exhibits a coefficient that is 0.04 higher than that of the B4 model, representing a substantial enhancement of 55.6% (see Supplementary Information **Fig. S.3c**).

Efficiency assessments suggest that the 3D-SC design not only boosts thrust but also diminishes power consumption, thus enhancing propulsive efficiency. When assessed at an equivalent thrust of 50 gf, the 3D-SC prototype exhibits a reduction in mechanical power consumption by 0.17 W, an improvement of 4.1% compared to the B1 design, indicating the marginal efficiency benefits conferred by 3D serrations. In contrast, the B1 versus B3 comparison highlights a significant efficiency leap with the cicada wing planform, achieving a 1.49 W power reduction, equating to a 26.7% decrease. Against the B4 design, the 3D-SC model exhibits a substantial 20.2% cut in power consumption, reinforcing its dual advancements in thrust production and energy savings. Confined by the limited strength of digital ABS material, the rotor diameter is set as 6 inches, allowing for the assessment of all prototypes across a wider spectrum of rotational speeds without causing significant elastic deformation. This dimension is notably smaller than the propellers commonly employed in drones³. To investigate the scalability of our findings for larger-scale applications, additional experiments are performed on 12-inch 3D-SC and B4 propeller prototypes, operating at a maximum rotational speed of 3000 RPM. The outcomes of these tests are detailed in **Fig. S.4**, enabling the assessment of propeller performance at elevated thrust levels. At an equivalent thrust of 150 gf, the 3D-SC propeller achieves a mechanical power reduction of 2.29 W, corresponding to an improvement of 22.6% over the B4 prototype.”

Updates in section **Supplementary Information**:

“Aerodynamic Testing Platform: We utilize the TYTO Series 1585 thrust stand for all aerodynamic tests, as shown in **Figure S8**. This device delivers real-time measurements of various parameters, including the rotor’s electrical power input, torque, thrust, and rotational speed, with a uniform sampling rate of 40 Hz. To improve the data accuracy, each thrust and mechanical power curve presented in the manuscript comprises of 100 data points. Mitigating measurement uncertainty further, each data point is taken by the average of 100 measurements. Notably, the measurement steadiness decreases as the rotational speed increases. Nevertheless, the maximum level of uncertainty is maintained within an acceptable range. Specifically, at 50 gf thrust, the coefficient of variation for thrust and torque measurements is recorded at 0.33% and 2.2% respectively. Owing to the employment of a TYTO optical RPM probe, the measurement error in rotational speed is negligible.”

Updated **Figures**:

Fig. S.8: Experimental setup for aerodynamic data collection.

Reviewer #3 (Remarks to the Author):

+++++++ SUMMARY ++++++

The authors have conducted a study with the objective of enhancing the performance of traditional propeller systems. Specifically, they have investigated the combined advantages of incorporating features from owl wings and cicada wings. Owl wing geometry has been leveraged to reduce noise, while cicada wing characteristics have been introduced to mitigate aerodynamic performance degradation. This research encompasses both experimental and computational approaches. The manuscript discusses the benefits of the proposed solution, focusing on improvements in aerodynamic efficiency and noise reduction, and supported by a selection of findings. Overall, the manuscript is well-structured and written in a clear manner.

+++++++ MAIN CRITICISMS ++++++

This reviewer encounters difficulty in connecting the present work with the current state of the art. Is this the first time in which the advantages of bio-inspired planforms have been combined with bio-inspired trailing edge serrations? What types of planforms have previously been employed in conjunction with serrated trailing edge (TE) propellers?

We appreciate this valuable feedback which raises important points regarding the integration of our work with current state-of-the-art propeller designs. We would like to clarify that, to the best of our knowledge, this represents the first instance where cicada wing inspired planforms have been synergized with owl feather inspired 3D serrations across the entire propeller surface, not merely at the trailing edge. This integration of 3D serrations with a cicada planform is a novel approach in the field of propeller design.

While there is no universally established conventional planform (i.e., baseline line design) for propellers that incorporate trailing or leading-edge serrations^{1,2,5-8}, the conventional benchmark planform used in our study is derived from previous studies^{1,2}. Although a common standard has not been defined, these studies tend to position the maximum chord at the 20-25% span. This practice is recognized and accepted within the research community as a conventional baseline for comparative purposes.

Furthermore, we acknowledge that our selected conventional planform may not fully encapsulate the advancements of the latest state-of-the-art designs. In response, we have introduced a new benchmark, B4, to our study. This benchmark is a rescaled version of the DJI Phantom 3 drone propeller, esteemed for its performance and relevance as a contemporary industry standard. Through this addition, we aim to facilitate a better comparison between our innovative design and the latest practical applications. The revised experimental results and the ensuing discussions have been updated accordingly, which are presented below.

Updates in section **Introduction**:

“The comparison between B1 and B3 provides insights into the impact of the cicada planform. Furthermore, we introduce the DJI Phantom 3 propeller as an additional benchmark (B4), which serves as a reference for current state-of-the-art industry standards.”

Updates in section **Acoustic advancement at various rotational speeds in experiments** updates:

“Acoustic measurements of the 3D-SC and other benchmark prototypes are conducted at two distinct radial distances - 0.1 meters and 5 meters from the rotor - providing a near-field measurement and a relatively more distant comparative measurement. At each radial position, the microphone is circumferentially positioned around the thrust stand to collect sound data from different angles. In the experiment, the sound spectra of each prototype, corresponding to different thrusts and radial distances, are recorded from 0 to 2 kHz. The sound spectrum is subsequently integrated to evaluate OASPL, as shown in **Fig. 2.b**. These measurements are taken with the

microphone positioned directly in front of the rotor (0°). The trend indicates that the OASPL of rotor noise generally increases with thrust and decreases with the radial distance. Experimental results show the cicada wing inspired B1 prototype achieves up to 1.6 dB lower OASPL than the conventional B3 prototype at 0.1 meters with 50 gram-forces (gf) thrust, reducing to 1.4 dB at 5 meters. By contrast, the implementation of 3D serrations carries out a more significant reduction in noise. The 3D-SC model showcases a noise decrease of 8.5 dB relative to the B1 model under identical conditions (0.1 m, 50 gf). This effect of 3D serrations is consistently observed in its modification to the conventional planform. In particular, the OASPL of B2 is 8.8 dB lower than that of B3. In the context of industry benchmarks, the OASPL of the B4 prototype is reduced in comparison to the non-serrated models (B1 and B3), yet it remains higher than that of the serrated designs (3D-SC and B2). Notably, the 3D-SC prototype demonstrates a reduction of 5.0 dB when benchmarked against B4 at 5 m, 50 gf.

Fig. 2c presents the sound distribution in a polar coordinate system at distances of 0.1 m and 5 m, with radial and angular coordinates representing the OASPL and the measurement angle, respectively. A constant thrust of 50 gf is applied in this evaluation to simulate a high-thrust scenario akin to that of a drone propeller during flight. The level of noise reduction is more pronounced at the 0.1 m measurements. As the measurement distance increases, the amount of reduction declines accordingly. For a quantitative assessment of acoustic performance, a measurement distance of 5 meters is selected to emphasize drone noise as perceived in the far-field. The peak noise abatement is achieved at a 30° measurement angle, where the 3D-SC design demonstrates a 5.5 dB noise reduction compared to the industry benchmark B4. The analyses herein utilize the unweighted OASPL. However, for industry relevance, the A-weighted OASPL—which accentuates frequencies within the human auditory range—is often preferred. When examining frequencies critical to human hearing, the 3D-SC prototype delivers a maximum noise reduction of 3.3 dBA compared to the B4 design at a 30° measurement angle (see Supplementary Information **Fig. S.3a-b**).

To explore the effective noise attenuation bandwidth, an analysis of the rotor sound spectrum is conducted. Sound spectra at 15 gf (corresponding to $Re \approx 1.01e4$) and 50 gf (corresponding to $Re \approx 1.82e4$) are chosen to exemplify the acoustic characteristics of the propellers across different flow regimes, as depicted in **Fig. 2d**. The spectral analysis employs a 5th-order Savitzky-Golay filter²⁵, which was employed to smooth the data and reduce noise without significantly distorting the signal. The rotor spectrum at 15 gf is dominated by acoustic tones whose frequencies are lower than 10 kHz. This confirms that loading noise and its harmonics are the primary component of propeller noise at low rotational speeds. Notably, the SPLs of multiple tones of a 3D-SC propeller are apparently lower than B1-4. At higher Reynolds numbers represented by 50 gf, these SPL peaks diminish in prominence, giving way to broadband noise as the primary acoustic signature. A comparative study of the B1 and B3 prototypes reveals that the cicada wing-inspired planform alone has a limited effect on noise mitigation, increasing peak SPL within the 5-9 kHz frequency band while reducing noise elsewhere. In contrast, the implementation of 3D serrations results in a SPL reduction across the entire spectrum. These experimental results of rotor noise across various thrusts and locations finds that the 3D-SC design greatly reduces noise levels compared to other benchmarks.”

Updates in section **Experimental evidence of 3D-SC aerodynamic advantages:**

“In terms of aerodynamic efficiency, the mechanical power, $\mathcal{W} = Q \cdot n$, of a propeller is a critical determinant of energy consumption rates during flight, where Q is the propeller torque.

As depicted in **Fig. 2e**, the thrust generation of propellers is assessed across the rotational speeds up to 6000 RPM, with average mechanical power measured at thrust levels from 0 to 50 gf. The 3D-SC design outperforms other models across all speeds, notably at higher RPMs. A notable thrust enhancement of 14.8 gf, or 39.2%, is observed in the B1 versus B3 comparison at 5000 RPM, credited to the aerodynamic refinement inherent in the

cicada wing planform. At the same RPM, the 3D-SC design produces an extra 20.3 gf of thrust over B4. Analysis of the thrust coefficient reveals that the 3D-SC prototype exhibits a coefficient that is 0.04 higher than that of the B4 model, representing a substantial enhancement of 55.6% (see Supplementary Information **Fig. S.3c**).

Efficiency assessments suggest that the 3D-SC design not only boosts thrust but also diminishes power consumption, thus enhancing propulsive efficiency. When assessed at an equivalent thrust of 50 gf, the 3D-SC prototype exhibits a reduction in mechanical power consumption by 0.17 W, an improvement of 4.1% compared to the B1 design, indicating the marginal efficiency benefits conferred by 3D serrations. In contrast, the B1 versus B3 comparison highlights a significant efficiency leap with the cicada wing planform, achieving a 1.49 W power reduction, equating to a 26.7% decrease. Against the B4 design, the 3D-SC model exhibits a substantial 20.2% cut in power consumption, reinforcing its dual advancements in thrust production and energy savings. Confined by the limited strength of digital ABS material, the rotor diameter is set as 6 inches, allowing for the assessment of all prototypes across a wider spectrum of rotational speeds without causing significant elastic deformation. This dimension is notably smaller than the propellers commonly employed in drones³. To investigate the scalability of our findings for larger-scale applications, additional experiments are performed on 12-inch 3D-SC and B4 propeller prototypes, operating at a maximum rotational speed of 3000 RPM. The outcomes of these tests are detailed in **Fig. S.4**, enabling the assessment of propeller performance at elevated thrust levels. At an equivalent thrust of 150 gf, the 3D-SC propeller achieves a mechanical power reduction of 2.29 W, corresponding to an improvement of 22.6% over the B4 prototype.”

Updates in **Supplementary Information**:

“Aerodynamic Testing Platform: We utilize the TYTO Series 1585 thrust stand for all aerodynamic tests, as shown in **Figure S8**. This device delivers real-time measurements of various parameters, including the rotor’s electrical power input, torque, thrust, and rotational speed, with a uniform sampling rate of 40 Hz. To improve the data accuracy, each thrust and mechanical power curve presented in the manuscript comprises of 100 data points. Mitigating measurement uncertainty further, each data point is taken by the average of 100 measurements. Notably, the measurement steadiness decreases as the rotational speed increases. Nevertheless, the maximum level of uncertainty is maintained within an acceptable range. Specifically, at 50 gf thrust, the coefficient of variation for thrust and torque measurements is recorded at 0.33% and 2.2% respectively. Owing to the employment of a TYTO optical RPM probe, the measurement error in rotational speed is negligible.”

Updated **Figures**:

Fig. 4. Illustration of the design concepts for a 3D-SC propeller. (a) 3D-SC topology inspired by owl feather and cicada forewing geometry. (b) Fabricated 3D-SC, B1, B2, B3, and B4 prototypes using Polyjet additive manufacturing

Fig. 5. Aerodynamic and aeroacoustic experimental results. (a) CAD schematics of 3D-SC, B1, B2, B3, and B4 propeller designs with associated color codes. (b) OASPL data across thrusts ranging from 10 to 50 gf. (c) Sound spectra comparison at 0.1 m (top) and 5 m (bottom) radial distances under 50 gf thrust conditions. (d) Polar representation of OASPLs at different measurement angles, with data collected at radial distances of 0.1 m (left) and 5 m (right). (e) Graphs depicting the relationship between thrust and rotational speed (upper) and mechanical power versus thrust (lower).

Fig. S.3: Supplementary experimental results. (a) A-weighted OASPL across different thrust levels. (b) Polar representation of A-weighted OASPL at varying measurement angles. (c) Thrust coefficient against rotational speed.

Fig. S.8: Experimental setup for aerodynamic data collection.

In the opinion of this reviewer, the followed methodology does not allow to directly conclude on the benefits of the proposed technology, and the work could be considered a preliminary study. It remains unclear what the optimal operating parameters are for each of the proposed designs, and e.g. no consideration was given to pitch variation. Additionally, the intended application for this technology is not clear to this reviewer. Factors such as scale effects, manufacturability, and durability would play a pivotal role when translating the findings of this study into industrial applications.

We thank the reviewer for the discerning observation regarding pitch angle variation. This comment has accurately identified that our study did not delve deeply into the optimization of pitch angles for each chord line. The primary reason for this is the current lack of knowledge regarding the optimal pitch angle configuration within such a novel design framework. Although the cicada wing's morphology provides a straightforward model for determining the chord lengths of 2D airfoil profiles, it does not extend to inform the aerodynamic design of pitch variations. Pitch variation, while crucial for aerodynamic efficiency, stands distinct from the biological mimicry that underpins our design's conception. As such, research dedicated to understanding and optimizing pitch angle effects—an endeavor that is strictly aerodynamic—falls outside the immediate scope of our bio-inspiration-focused study.

Nevertheless, the reviewer's insight has brought to our attention that the conventional planform features a variable pitch angle from root to tip, a difference when juxtaposed with the cicada planform which maintains a constant pitch angle throughout its span. To ensure a better comparison, we have redesigned the B3 and B2 models. The revised acoustic and aerodynamic test outcomes are detailed in our response to the reviewer's above previous comment. Furthermore, we acknowledge the importance of the reviewer's comments on scale effects, manufacturability, and durability. This study includes an initial assessment of our design's scalability, achieved by increasing the prototype propellers' diameter to 12 inches. The early findings from this scalability test are documented in the Supplementary Information and are summarized below. In our future research endeavors, we will aim to incorporate these additional factors into the development process.

Experimental evidence of 3D-SC aerodynamic advantages modifications:

“To investigate the scalability of our findings for larger-scale applications, additional experiments are performed on 12-inch 3D-SC and B4 propeller prototypes, operating at a maximum rotational speed of 3000 RPM. The outcomes of these tests are detailed in **Fig. S.4**, enabling the assessment of propeller performance at elevated thrust levels. At an equivalent thrust of 150 gf, the 3D-SC propeller achieves a mechanical power reduction of 2.29 W, corresponding to an improvement of 22.6% over the B4 prototype.”

Updated **Fig. S.4**.

Fig. S.4, Comparative aerodynamic analysis of 12-inch 3D-SC and B4 prototype models. (a) The 3D-printed prototypes featuring the 12-inch model above and the 6-inch model below. (b) Thrust versus rotational speed plot (left) and mechanical power against thrust plot (right).

There seems to be no information on how the authors dealt with the uncertainties of their experiments.

We thank the reviewer for this valuable comment. We acknowledge that the measurement of propulsive efficiency pose challenges in terms of stability and introduced additional noise, particularly through the current and voltage measurements. To mitigate this and improve the precision of our data, we have undertaken significant improvements to our aerodynamic testing apparatus.

These enhancements have been specifically designed to yield more stable measurements and to reduce the uncertainty associated with quantifying a propeller's mechanical power consumption. As a result, we now achieve a more accurate and direct assessment of the energy consumption rate of a propeller under a fixed thrust condition. Furthermore, we have expanded our methodology to include acoustic measurements at various distances, angles, and thrust levels, ensuring a thorough and impartial evaluation of the aerodynamic and acoustic behaviors of the 3D-SC topology.

We have revised our manuscript to include a detailed description of these apparatus upgrades, the methodology for handling uncertainties, and the statistical analysis applied to our data. The updated results and discussions, which reflect these enhancements, are presented in our responses to the reviewer's earlier comments above.

+++++ OTHER COMMENTS +++++

[Abstract and Introduction] The translation of bio-inspired shapes into the geometrical adaptation of the propeller could be made clearer. Specifically, the utilization of owl wings as inspiration for defining TE serrations and cicada wings for the planform should be explicitly explained.

We thank the reviewer for bringing this to our attention. We would like to clarify the serration topology featured in our research. The design we have introduced does not pertain to traditional trailing edge (TE) serrations but represents an innovative 3D serration affecting the propeller's entire surface, a departure from the typical TE approach. In response to this comment, we have revised the introduction of our manuscript. These sections now clearly differentiate our 3D serration design from traditional TE serrations and more thoroughly explain the bio-inspirations for our serrations and planform.

Updates in **Introduction**:

“Inspired by the morphology of owl feathers, our design introduces a high-fidelity, three-dimensional (3D) sinusoidal serration topography that encompasses a widespread surface adaptation rather than a localized edge variation for potential acoustic improvement, as illustrated in **Fig. 1a**. Integrating this design, the cicada-inspired wing shape serves as the foundational planform for the propeller to augment aerodynamic efficiency.”

We have also updated the title of the manuscript to better describe the translation of bioinspiration into geometrical adaptation: Owl-inspired serrations merge with cicada-derived planform: A synergistic blueprint for silent and efficient flight.

[Page 1] This reviewer has difficulty understanding the choice of using B-splines as a constraint for design. In the present study, the authors employed trigonometric functions and polynomials to represent their geometry, which are arguably more restrictive than B-splines.

We thank the reviewer for this comment. We acknowledge that B-splines offer significant flexibility in design due to their capacity to manipulate a large number of control points. However, our choice to employ trigonometric functions and polynomials was dictated by the specific nature of the bio-inspired cicada planform, which is a design approach distinct from that of B-splines. The cicada planform we have adopted is derived from a fundamentally different design philosophy, one that is tailored to replicate the intricate patterns found in nature. As for the 3D serration texture, it is a separate feature that complements the planform and can be applied over any planform, including those designed with B-splines. This additive texture is not intended to compete with, but rather to enhance, the underlying airfoil design, regardless of the method used for its creation.

We have revised the relevant section of our manuscript to clearly articulate the rationale behind our design approach and to delineate the distinct roles of the cicada planform and the 3D serration texture. The updated text is presented below.

Updates in **Introduction**:

“In the conventional framework of propeller design, B-spline methodologies¹ play a pivotal role. This involves generating a series of control points through basis functions, which are instrumental in formulating the aerodynamic surface. Nevertheless, the interplay between aerodynamic efficiency and noise reduction is delicate, and finding a balance often requires stepping outside established design paradigms. Researchers are alternatively exploring the flight mechanisms of natural predators renowned for their stealth, aiming to derive innovative and efficacious design principles.”

[Page 3] How did the authors design the serrations and the cicada planform? Was it based on the study and scaling of a single individual, or did it result from statistical analysis?

We thank the reviewer for bringing this to our attention. The design of 3D serrations is discussed in section **Topological design concepts**. With respect to the cicada planform, we have based our design on the scaling of a single cicada individual. We have included additional information in the manuscript to clarify this design method, which is detailed in the updated section below.

“Notably, the spline functions defining the leading and trailing edges are derived directly from the cicada wing's outline, as depicted in **Fig. 1a**. The selection of the cicada planform currently serves as a baseline representation rather than an optimized solution. Optimization of this topology for advanced performance parameters remains a prospective avenue for extended research beyond the scope of this study.”

[Page 4] Why did the authors consider the effective angle of attack in Equation 6? This reviewer is not familiar with such an approach. Does this definition align with the information provided in the supplementary material?

We thank the reviewer for sharing this comment with us. To address this question, we have developed a schematic illustration that outlines the derivation process in a visual format shown below:

To enhance comprehension of this work, certain essential information found in the supplementary material could be integrated into the main manuscript.

We thank the reviewer for this comment. We have included the essential descriptions on CFD and experimental methods in the main paper and put the other information (i.e., theoretical derivations) in supplementary material.

+++++++ FORMAL ASPECTS ++++++

On page 10, the "Conclusions" should be given a dedicated section, as they currently appear as a paragraph within the results section.

We thank the reviewer for this comment. We didn't include a dedicated section entitled "Conclusions" because of the journal formatting requirements.

References

- 1 Priatmoko, M. & Nirbito, W. in IOP Conference Series: Materials Science and Engineering. 012008 (IOP Publishing).
- 2 Lee, H. M., Lu, Z., Lim, K. M., Xie, J. & Lee, H. P. Quieter propeller with serrated trailing edge. *Applied Acoustics* 146, 227-236 (2019).
- 3 Wan, H., Dong, H. & Gai, K. Computational investigation of cicada aerodynamics in forward flight. *Journal of The Royal Society Interface* 12, 20141116 (2015).
- 4 Lilley, G. in 4th AIAA/CEAS aeroacoustics conference. 2340.
- 5 Ning, Z. & Hu, H. in 35th AIAA Applied Aerodynamics Conference. 3747.
- 6 Lan, T., Li, G. & Zhang, M. in *Journal of Physics: Conference Series*. 012012 (IOP Publishing).
- 7 Wei, Y., Qian, Y., Bian, S., Xu, F. & Kong, D. Experimental study of the performance of a propeller with trailing-edge serrations. *Acoustics Australia* 49, 305-316 (2021).
- 8 Wei, Y., Xu, F., Bian, S. & Kong, D. Noise reduction of UAV using biomimetic propellers with varied morphologies leading-edge serration. *Journal of Bionic Engineering* 17, 767-779 (2020).

REVIEWER COMMENTS

Reviewer #1 (Remarks to the Author):

Thank you to the authors for their thorough adjustments to the manuscript and experimental approaches. The resulting manuscript does provide an important contribution to the field and is an exciting result and I do believe it deserves to be published. However there are a few more critical items that should be addressed prior to publication. In particular, the lack of uncertainty/statistical discussions must be rectified.

Major items:

As highlighted by another reviewer, measurement uncertainty is missing throughout the text and, although the authors state statistical analysis has now been completed, it is not clear to this reviewer where in the main text or supplementary materials those statistical analyses are. The authors state the uncertainty is within an acceptable range, what is that and how was that selected? Results in the text are provided as average single values or percentages with no standard deviation/error or discussion of the repeatability of the experiment. Furthermore, error bars are missing from all experimental results and should be included to determine if differences between approaches are statistically significant. A proper experimental uncertainty analysis with statistics will be critical for publication.

Although I now feel convinced after the author's review response, the manuscript itself struggles to help a reader connect the logic of why one would combine owl and cicada features. To help make this connection, I have additional wording suggestions below.

This paper is a good example of when engineering has enhanced beyond bio-inspired approaches as evidenced in the section where the authors identified that the leading-edge serrations are noisier than their proposed 3D surface serrations. This result feels like a side point in the current framing, but it holds substantial value.

Page 7. It is not clear why a CFD methods paragraph is the lead in paragraph for the discussion, and the paragraphs that follow read as additional results. There is effectively only one paragraph that would be a traditional "discussion section" before transitioning into the final summary paragraph.

Minor items:

Abstract. The first two sentences are only about owls followed by introducing this experiment as a mix of owl and cicadas. I think this sells the work short. I would recommend highlighting that multiple animals have beneficial flight adaptations and that your group thinks these differences could be combined for beneficial outcomes or something that helps a reader follow your logic.

Page 2. In response to this reviewer's previous minor item about Reynolds number: Although I understand the author's logic now, it remains important for the reader to understand that the function of biological characteristics is linked to the environment in which the animal operates. Given the authors' reply, this issue could be nicely addressed in the introduction by including a sentence that acknowledges the difference of Reynolds number between owls and cicadas and directly includes your response of "Despite these differences, existing research suggests that cicada wing planforms offer aerodynamic benefits across a wide range of Re , including those at which convention drone props operate⁵." A sentence along those lines will highlight important logic behind your study from a reader's perspective that is otherwise unclear.

Page 3. Broken link to Eq. 5

Reviewer #3 (Remarks to the Author):

From the point of view of this reviewer, the authors have addressed all the points raised in the initial revision. Their letter shows detailed responses, and the outcome of the discussion has been smoothly incorporated into the manuscript or the supplementary data.

REVIEWER COMMENTS

We thank the reviewers for their insightful comments and critique of the presented work. Please find below a point-by-point response to every comment and indications of what and where changes have been made. These changes are indicated in blue text in the revised paper.

Reviewer #1

Thank you to the authors for their thorough adjustments to the manuscript and experimental approaches. The resulting manuscript does provide an important contribution to the field and is an exciting result and I do believe it deserves to be published. However, there are a few more critical items that should be addressed prior to publication. In particular, the lack of uncertainty/statistical discussions must be rectified.

Major items:

As highlighted by another reviewer, measurement uncertainty is missing throughout the text and, although the authors state statistical analysis has now been completed, it is not clear to this reviewer where in the main text or supplementary materials those statistical analyses are. The authors state the uncertainty is within an acceptable range, what is that and how was that selected? Results in the text are provided as average single values or percentages with no standard deviation/error or discussion of the repeatability of the experiment. Furthermore, error bars are missing from all experimental results and should be included to determine if differences between approaches are statistically significant. A proper experimental uncertainty analysis with statistics will be critical for publication.

We thank the reviewer for this valuable comment. As suggested by the reviewer, we have added additional discussion of the data uncertainty in our manuscript and supplementary information. Furthermore, we conducted additional experiments and have added error bars to the plots that are related to our acoustic tests. Below includes discussion and figures on the data uncertainty and repeatability of our experiments:

“Acoustic advancement at various experimental conditions

Regarding data uncertainty, the microphone positioned at 0.1 meters recorded a standard deviation in OASPL ranging between 0.48 and 0.77 dB, with a maximum coefficient of variation (CV) of 6.6%. At 5 meters, the standard deviation fluctuated between 0.41 and 0.72 dB, with the maximum CV reaching 7.9%.

Experimental evidence of 3D-SC aerodynamic advantages

Thrust measurement variability is characterized by standard deviations ranging from 0.29 gf to 0.66 gf, with a maximum CV of 5.34% at the 10 gf thrust level and 1.0% at 50 gf. Concerning mechanical power measurements, the predominant source of uncertainty originated from torque acquisition. Power fluctuations had a standard deviation between 0.032 to 0.097 W, yielding a maximum CV of 12 % at 10 gf and 1.7% at 50 gf. A detailed quantification of the aerodynamic data uncertainty across various thrust levels is delineated in Fig. S.4.

Supplementary Information – Aerodynamic testing platform

Regarding the measurement uncertainty, the fluctuation in rotational speed is negligible due to the use of a TYTO optical RPM probe. To accurately determine uncertainty in thrust and power measurements, we

collected 1000 data points at each thrust level. The statistical distribution of these measurements is depicted in boxplots in Fig. S.4.”

Fig. 1. Aerodynamic and aeroacoustic experimental results. (a) CAD schematics of 3D-SC, B1, B2, B3, and B4 propeller designs with associated color codes. (b) OASPL data across thrusts ranging from 10 to 50 gf. (c) Sound spectra comparison at 0.1 m (top) and 5 m (bottom) radial distances under 50 gf thrust conditions. (d) Polar representation of OASPLs at different measurement angles, with data collected at radial distances of 0.1 m (left) and 5 m (right). (e) Graphs depicting the relationship between thrust and rotational speed (upper) and mechanical power versus thrust (lower).

Fig. S.3. Supplementary experimental results. (a) A-weighted OASPL across different thrust levels. (b) Polar representation of A-weighted OASPL at varying measurement angles. (c) Thrust coefficient against rotational speed.

Fig. S.4. Assessment of data uncertainty for (a) thrust and (b) mechanical power measurements across various thrust levels.

Although I now feel convinced after the author's review response, the manuscript itself struggles to help a reader connect the logic of why one would combine owl and cicada features. To help make this connection, I have additional wording suggestions below.

This paper is a good example of when engineering has enhanced beyond bio-inspired approaches as evidenced in the section where the authors identified that the leading-edge serrations are noisier than their proposed 3D surface serrations. This result feels like a side point in the current framing, but it holds substantial value.

We thank the reviewer for this insightful feedback. We agree that this suggestion can help clarify our original motivation of combining owl and cicada wing morphology. We have made the following modifications to the updated **Discussion** and **Acoustic comparison with 2D leading-edge serrations** section, as shown below:

“Acoustic comparison with 2D leading-edge serrations:

This empirical result indicates that the extended surface texture of 3D serrations plays a pivotal role in achieving a lower acoustic emission compared to 2D serrations, thus marking a significant development in the serration-based noise suppression strategies.

Discussion:

In summary, this work on 3D-SC surface topologies, inspired from the morphological traits of cicadas and owls, has demonstrated a reduction in propeller noise across the frequency spectrum alongside improvements in aerodynamic efficiency compared to benchmark designs. The implementation of 3D serrations yields more substantial noise reduction beyond that of 2D counterparts due to the extended surface textures. Furthermore, comparative analyses with benchmark designs have demonstrated that incorporating a cicada wing planform notably augments thrust and propulsive efficiency. Despite the intrinsic difference in the operating Re characterizing the flight of owls and cicadas, the amalgamation of these two distinct morphologies leads to a concurrent enhancement of aerodynamic efficiency and noise suppression. We would like to highlight that such improvements are unattainable through either geometric feature in isolation. They are uniquely a result of the synergistic integration of these morphological elements.”

Page 7. It is not clear why a CFD methods paragraph is the lead in paragraph for the discussion, and the paragraphs that follow read as additional results. There is effectively only one paragraph that would be a traditional “discussion section” before transitioning into the final summary paragraph.

We are grateful to the reviewer for the insightful feedback. In response, we have restructured the manuscript to create a distinct section titled "**Analysis of vortex manipulation mechanisms for rotor noise mitigation.**" This section is dedicated to elucidating the CFD methods and the pertinent findings. Following this addition, the concluding part of the paper has been refined into a **Discussion** section, which encapsulates the main points of our research.

Minor items:

Abstract. The first two sentences are only about owls followed by introducing this experiment as a mix of owl and cicadas. I think this sells the work short. I would recommend highlighting that multiple animals have beneficial flight adaptations and that your group thinks these differences could be combined for beneficial outcomes or something that helps a reader follow your logic.

We thank the reviewer for bringing this to our attention. We have modified the abstract to better emphasize on the synergistic design strategy as the reviewer suggested, as shown below.

“As natural predators, owls fly with astonishing stealth due to the serrated feather morphology that produces advantageous flow characteristics. Traditionally, these serrations are tailored for airfoil edges with simple two-dimensional patterns, limiting their effect on noise reduction while negotiating tradeoffs in aerodynamic performance. Conversely, the intricately structured wings of cicadas have evolved for effective flapping, presenting a potential blueprint for alleviating these aerodynamic limitations. In this study, we formulate a synergistic design strategy that harmonizes noise suppression with aerodynamic efficiency by integrating the geometrical attributes of owl feathers and cicada forewings, culminating in a three-dimensional propeller topology that facilitates both silent and efficient flight.”

Page 2. In response to this reviewer’s previous minor item about Reynolds number: Although I understand the author’s logic now, it remains important for the reader to understand that the function of biological characteristics is linked to the environment in which the animal operates. Given the authors’ reply, this issue could be nicely addressed in the introduction by including a sentence that acknowledges the difference of Reynolds number between owls and cicadas and directly includes your response of “Despite these differences, existing research suggests that cicada wing planforms offer aerodynamic benefits across a wide range of Re , including those at which convention drone props operate⁵.” A sentence along those lines will highlight important logic behind your study from a reader’s perspective that is otherwise unclear.

We thank the reviewer for this valuable comment, we have added the following to the introduction section:

“It is noteworthy that there exists a variation in Reynolds number (Re) between the flight of owls¹ ($Re \approx 6e4$) and cicadas² ($Re \approx 2.0e3$). Despite these differences, existing research from numerical simulations²⁻⁴ and empirical studies⁵ suggests that cicada wing planforms offer aerodynamic benefits across a wide range of Re , including those at which conventional drone propellers operate^{6,7}.”

Page 3. Broken link to Eq. 5

We thank the reviewer for pointing this out. We have fixed the broken link to Eq. (5).

Reviewer #3:

From the point of view of this reviewer, the authors have addressed all the points raised in the initial revision. Their letter shows detailed responses, and the outcome of the discussion has been smoothly incorporated into the manuscript or the supplementary data.

We deeply appreciate the reviewer for reviewing our paper and the insightful comments. Thank you!

References

- 1 Wagner, H., Weger, M., Klaas, M. & Schröder, W. Features of owl wings that promote silent flight. *Interface focus* **7**, 20160078 (2017).
- 2 Wan, H., Dong, H. & Gai, K. Computational investigation of cicada aerodynamics in forward flight. *Journal of The Royal Society Interface* **12**, 20141116 (2015).
- 3 Liu, G., Dong, H. & Li, C. Vortex dynamics and new lift enhancement mechanism of wing–body interaction in insect forward flight. *Journal of Fluid Mechanics* **795**, 634-651 (2016).

- 4 Birch, J. M. & Dickinson, M. H. Spanwise flow and the attachment of the leading-edge vortex on insect wings. *Nature* **412**, 729-733 (2001).
- 5 Tsuyuki, K., Sudo, S. & Tani, J. Morphology of insect wings and airflow produced by flapping insects. *Journal of intelligent material systems and structures* **17**, 743-751 (2006).
- 6 Hintz, C., Khanbolouki, P., Perez, A. M., Tehrani, M. & Poroseva, S. in *2018 Applied Aerodynamics Conference*. 3645.
- 7 Ning, Z. & Hu, H. in *35th AIAA Applied Aerodynamics Conference*. 3747.

REVIEWERS' COMMENTS

Reviewer #1 (Remarks to the Author):

All points raised have been addressed. Thank you to the authors for thoughtfully incorporating the review suggestions at all stages of this process. I would like to congratulate the team on a great paper.